# GelMap: intrinsic calibration and deformation mapping for expansion microscopy

Hugo G. J. Damstra [1,4], Josiah B. Passmore [1,2,4], Albert K. Serweta[1], Ioannis Koutlas [3], Mithila Burute[1], Frank J. Meye[3], Anna Akhmanova [1] & Lukas C. Kapitein [1,2]✉

Expansion microscopy (ExM) is a powerful technique to overcome the diffraction limit of light microscopy by physically expanding biological specimen in three dimensions. Nonetheless, using ExM for quantitative or diagnostic applications requires robust quality control methods to precisely determine expansion factors and to map deformations due to anisotropic expansion. Here we present GelMap, a flexible workflow to introduce a fluorescent grid into pre-expanded hydrogels that scales with expansion and reports deformations. We demonstrate that GelMap can be used to precisely determine the local expansion factor and to correct for deformations without the use of cellular reference structures or pre-expansion ground-truth images. Moreover, we show that GelMap aids sample navigation for correlative uses of expansion microscopy. Finally, we show that GelMap is compatible with expansion of tissue and can be readily implemented as a quality control step into existing ExM workflows.

In ExM the effective resolution of light microscopy is increased by physical expansion of cells and tissues[1]. Biological specimens are anchored to a swellable hydrogel, typically by conjugation of an acrylate group to free lysines and subsequent polymerization of an acrylamide gel that includes sodium acrylate. Next, the sample is chemically homogenized to detach the gel from the culturing substrate and prevent resistance during expansion. Finally, addition of water induces swelling in all dimensions[2]. As ExM can be used to directly expand cells and intact tissues, ExM has quickly become an important technique in biological research that also has great potential for diagnostic purposes in clinical settings[3,4].

In recent years, various specialized ExM variants have been described, focusing on improving preservation of specific structures[5], expanding tissues and entire organisms[6–9], including human biopsy tissues[3,4], and compatibility with other super-resolution modalities[10–13]. As the effective resolution of ExM is in part limited by the expansion

factor, there has also been a push to develop higher expanding methods compared to the original ExM method either by varying the crosslinking concentration[1,14], using alternative crosslinking chemistry[15,16] or by using iterative approaches[17,18]. The push for higher expansion and more complex ExM applications means that robust characterization of the exact expansion factor and of possible local deformations will be essential for biological reproducibility, quantitative accuracy and diagnostic applications.

In most cases, the expansion factor of an imaged sample is determined macroscopically by measuring the size of the expanded gel. In addition, ExM variants can be validated for a microscopic expansion factor using subcellular reference structures, such as nuclear pore complexes or clathrin-coated pits, which have a distinct size known from other modalities such as electron microscopy[14,19]. Alternatively, the precise local expansion factor can be determined by correlating a larger reference structure, for example, a stained cell, before expansion

[1]Cell Biology, Neurobiology and Biophysics, Department of Biology, Faculty of Science, Utrecht University, Utrecht, The Netherlands. [2]Centre for Living Technologies, Alliance TU/e, WUR, UU, UMC Utrecht, Utrecht, The Netherlands. [3]Department of Translational Neuroscience, Brain Center, UMC Utrecht, Utrecht University, Utrecht, The Netherlands. [4]These authors contributed equally: Hugo G. J. Damstra, Josiah B. Passmore. ✉e-mail: l.kapitein@uu.nl

to the same structure after expansion. This is a powerful approach as it can also be used to map local deformations that occur during expansion and sample mounting; however, due to the physical differences between pre- and post-expanded samples, robust correlation is time consuming, laborious, relies on efficient sample navigation and efficient labeling of the structure of interest, and is low throughput, making it challenging to repeatedly assess reproducibility and variability in terms of expansion and deformation for different gels and samples. Moreover, there are cases when pre- and post-expansion correlation is not possible, for example in the case of post-expansion labeling, where the structure of interest will not be visible before expansion. Current attempts to directly measure the microscopic expansion factor without using cellular reference structures or pre-expansion imaging are not readily compatible with biological samples and do not provide information about local deformations[20]. Therefore, easy-to-use reference-free quality control mechanisms are currently lacking, which hampers widescale adoption of ExM for quantitative purposes, including diagnostic applications.

Here, we introduce GelMap, a workflow wherein a gel-embedded reference grid is used to intrinsically calibrate hydrogels. GelMap functions as an expandable ruler that accurately reports (local) expansion factors, enables reference-free deformation mapping and facilitates sample navigation. This reference grid can be directly used as a culturing substrate or be incorporated into the gel during later steps of sample preparation. By primarily using protein photolithography to pattern GelMap grids onto coverslips, we demonstrate that depending on the experiment, different grid designs, patterned proteins or fluorescent groups can be incorporated into the expansion hydrogel together with any biological specimen to reliably report expansion factors and enable computational correction of local deformations. Additionally, GelMap also solves the challenge of sample navigation in expansion microscopy by directly incorporating a fluorescent coordinate system into the ExM hydrogel. This facilitates experiments in which live-cell imaging is followed up by expansion microscopy. Finally, we demonstrate that GelMap grids can be used beyond cultured cells by imprinting them during gelation, enabling calibration and deformation correction of expanded brain tissue slices. In summary, GelMap provides a key quality control step for expansion microscopy, ready for easy adoption in existing established ExM workflows in both research and clinical settings.

## Results

### Development of GelMap

Quantitative interpretation of expanded samples relies on robust characterization of expansion fidelity, which could be compromised by local anisotropies in expansion due to differences in gel composition, density or mounting. We reasoned that ExM hydrogels could be intrinsically calibrated by incorporating a scalable fluorescent fiducial pattern, which would allow for precise determination of the local microscopic expansion factor. By generating a culturing surface that contains a fluorescent protein-based pattern of repeating squares (Fig. 1a(i)) and directly culturing cells on patterned coverslips, both the cells and the pattern would be incorporated into the ExM hydrogel simultaneously (Fig. 1a(ii)). During expansion, the pattern will expand together with the cells and act as a scalable ruler to determine the expansion factor. As the marker consists of a repeating grid of known dimensions, local deformations that occur during expansion should be reflected in a visibly distorted grid (Fig. 1a(iii)) that could be used to computationally correct for deformation without relying on a pre-expanded reference state (Fig. 1a(iv)). To test this idea, we set out to develop a scalable fluorescent grid compatible with ExM chemistry that would be transferable from culture surface to hydrogel and biologically inert to ensure biological reproducibility and compatibility with all cell types.

Inspired by the use of deep-UV photolithography to create specific adhesion sites for cells by locally burning away a repulsive polyethylene

glycol layer[21], we reasoned that we could use photolithography to generate surfaces with well-defined protein patterns. To this end, we coated coverslips with fluorescent protein rather than polyethylene glycol and patterned the deep-UV illumination using a photomask. We designed various chrome–quartz photomasks to test different numbered grid designs, with repeated features ranging in size from 10 μm to 40 μm to focus on mapping deformations and larger grid sizes from 200 μm to 400 μm that included markers to aid sample navigation (Extended Data Fig. 1a,b). We first tested our patterning approach using coverslips coated with laminin, an extracellular matrix protein (ECM) conjugated to a fluorophore (Extended Data Fig. 2). We found we could successfully pattern entire coverslips (Fig. 1b(i)), with multiple designs (Extended Data Fig. 1c), resulting in an approach that has sufficient spatial resolution to pattern grids with micrometer scale. Next, we tested whether the fluorescent pattern could be transferred from the culturing substrate to a hydrogel by gelating and expanding a region in the middle of the coverslip using Ten-fold Robust Expansion (TREx) microscopy[14]. Imaging of the same coverslip after gelation confirmed the pattern was efficiently transferred from the substrate to the hydrogel (Fig. 1b(ii)). When the resulting gel was imaged after moderate expansion (3×) to facilitate imaging of the entire gel, we observed local deformations that distorted the uniformity of the pattern and that were particularly pronounced near the edge of the gel (Fig. 1b(iii)). By registering the deformed image to the pre-expansion pattern, the deformations could be corrected using nonlinear thin-plate spline transformation of the deformed image as shown previously[1,14,22] (Extended Data Fig. 3), which restored the uniformity of the pattern in the expanded gel (Fig. 1b(iv)). Finally, we tested whether our approach could be directly used as a culturing substrate by sterilizing a grid and growing cells directly on the protein patterned coverslip. Following fixation and immunostaining for tubulin and laminin to amplify the fluorescent grid, the sample was imaged and subsequently expanded. Pre-expansion imaging confirmed that cells grew irrespective of the underlying protein grid (Fig. 1c; pre-expansion). Comparison of pre- and post-expansion images confirmed that cells and the pattern were expanded together and by aligning the expanded image to a virtual reference grid the local deformations could be corrected (Fig. 1c).

In the described workflow, the pattern was amplified using antibody labeling (Extended Data Fig. 1d). Using the laminin pattern, we observed a small degree of labeling of cell-derived laminin when amplifying the pattern, which could hinder grid identification and deformation mapping. Therefore, we tested patterning efficiency using fibrinogen, a different ECM protein, as well as a nanobody against a nonbiological target (R2-myc-his, hereafter NBD) that contains an orthogonal myc-tag that can be used for amplification without intracellular background (Extended Data Fig. 1e). Following conjugation to a fluorophore, we found that the bulkier ECM proteins patterned as efficiently as the smaller globular nanobody and in both cases the sterilized patterns could be used as a culturing surface. Using NBD patterns, we confirmed there was no effect on cellular morphology of cells cultured on bare glass versus protein patterned coverslips by quantifying cellular area and circularity (Extended Data Fig. 4a,b). We also did not observe an effect on the macroscopic expansion factor of gels prepared on bare glass coverslips or on protein patterned coverslips (Extended Data Fig. 4c,d).

To examine the applicability of GelMap across different variants of ExM, we tested our approach using four commonly used approaches: pro-ExM[2], MAP[8], TREx[14] and pan-ExM[18], an iterative expansion approach (Extended Data Fig. 5). These methods use different homogenization approaches (enzymatic digestion and thermal denaturation) and different anchoring approaches (acryloyl-X SE or a combination of paraformaldehyde and acrylamide), as summarized in Extended Data Fig. 5b. Using TREx, we also compared pre- and post-expansion labeling of both cellular structures and amplification of the reference grid and found that post-expansion amplification of the pattern was

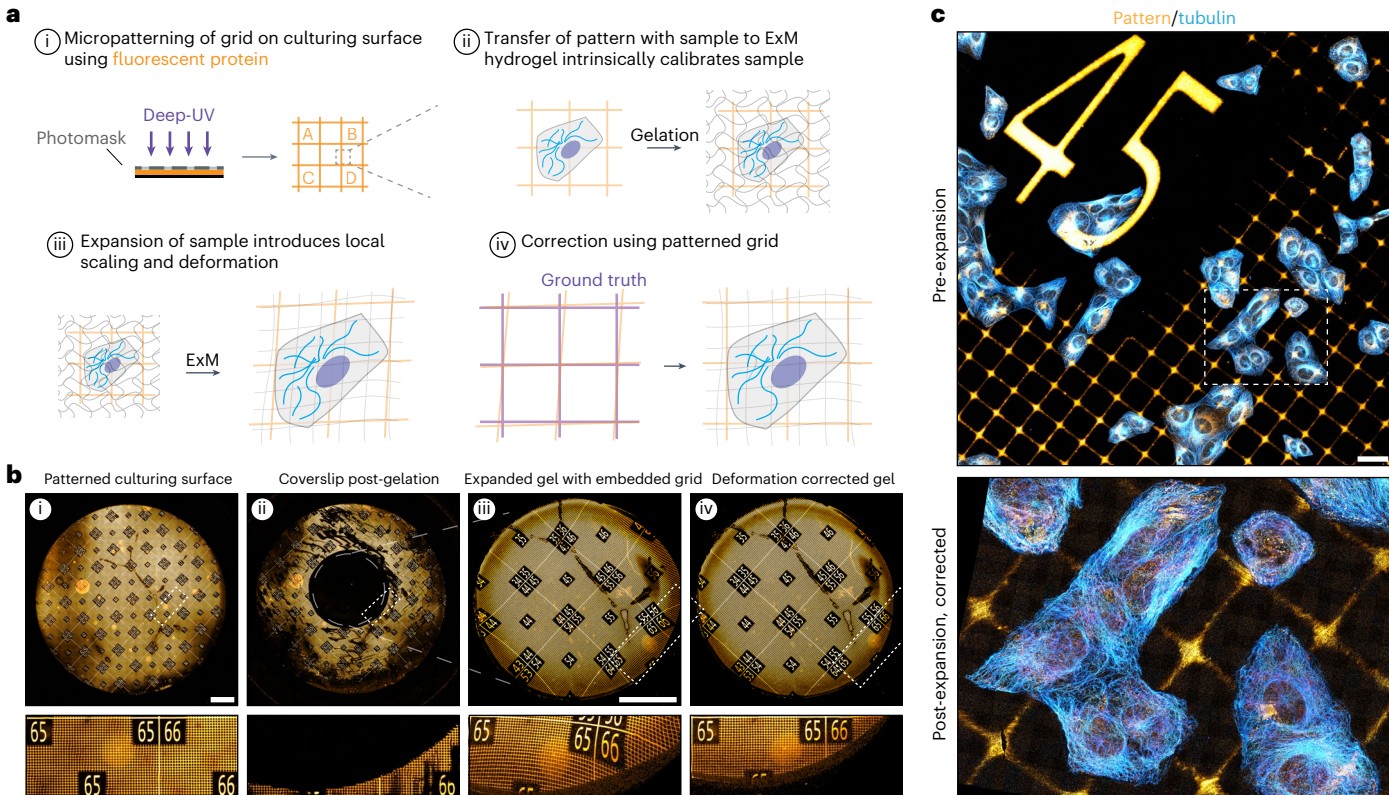

**Fig. 1 | GelMap enables intrinsic calibration and deformation mapping in expanded hydrogels. a**, Schematic of the general GelMap workflow. **b**, Representative images of the indicated steps of GelMap. (i) An 18-mm glass coverslip patterned with fluorescent laminin and amplified with antibody labeling (orange) shown before gelation. (ii) The same coverslip after digestion of a locally polymerized gel using a silicone mold in the middle of the coverslip (6 mm diameter), showing complete transfer of protein pattern from the coverslip. (iii) The polymerized and digested gel corresponding to the region indicated with gray circle in (ii) was expanded 3× and imaged, showing anisotropic deformation. (iv) The resulting acquisition was used for landmark-based registration to the pre-expanded image in (i) for correction of local deformations in the gel. **c**, Top: U2OS cells stained for tubulin (cyan) cultured directly on laminin-patterned GelMap coverslip and imaged before expansion. Bottom: Region marked in top image after expansion (expansion factor, 9.6×) using TREx, corrected using GelMap. Scale bars, 2 mm (**b**), zoomed regions 400 μm; 40 μm (**c**). Scale bars in expanded samples reflect pre-expansion sizes.

most efficient when used in combination with paraformaldehyde/acrylamide anchoring, but less when combined with acryloyl-X SE anchoring. Despite these differences, the reference grid was properly transferred to the expanded gel in all tested conditions and could be used to correct deformation, demonstrating that our approach can be easily adopted in a wide variety of expansion methods.

While the use of patterned coverslips is straightforward and robust, we also sought to establish an approach that could potentially enable patterning of three-dimensional (3D) grids within the polymerized hydrogel. To this end, we introduced an acrylate-modified photoactivatable rhodamine into the ExM gelation solution. After polymerization of the hydrogel, we introduced a pattern into the pre-expanded hydrogel by local photo-uncaging using targeted illumination. These photoactivated patterns expanded along with the ExM hydrogel (Extended Data Fig. 1f), albeit with a substantial degree of signal loss. In combination with patterned two-photon activation, a similar approach may facilitate a 3D patterning approach in future work.

Thus, our approach provides an easy-to-use, flexible method to intrinsically calibrate ExM hydrogels that is compatible with biological specimen and ExM chemistry. As this approach provides quantitative scaling information, reports on local deformations and can be used for easy navigation, we termed the resulting method GelMap.

## Correction of expansion anisotropy using GelMap
We next used GelMap to assess the expansion inhomogeneities and deformations that occur during expansion. We first examined how the estimated macroscopic expansion factor, as measured by the size of the expanded gel, relates to the measured microscopic expansion factor. We expanded GelMap grids without cells using TREx and compared the microscopic expansion factor of individual squares (two regions, 849 individual squares, expansion factor 9.0 ± 0.2 (mean ± s.d.)) with seven individual estimates of macroscopic expansion factor by unbiased participants. While the average measurement from the participants was consistent with the calculated average of the true expansion factor (9.0 ± 0.3, mean ± s.d.), the individual estimates ranged from 8.6 to 9.4. This highlights possible errors that can be introduced from individual measurements (Extended Data Fig. 6a). We next looked at the deformation of individual 40 × 40 μm squares within the same gel. We defined the squareness of each square within the field of view as a metric for ExM fidelity (accounting for both stretching and skewness) and observed marked variability across multiple gels and regions (three gels, three regions per gel, 3,574 individual squares) compared to the baseline error (determined by identifying crossings in resampled images of non-expanded grids) (Extended Data Fig. 6b). We could observe some regions that were hardly deformed (Fig. 2a; region 1) and other regions where very strong deformations were apparent, for example near tears (Extended Data Fig. 6c). We also noted more subtle anisotropies that became visible using GelMap, which may remain unrecognized when looking at biological structures alone (Fig. 2a; region 2). Confirming these qualitative observations, plotting the distribution of squareness values for both regions revealed more deformations in region 2 compared to region 1 (Fig. 2b). Together, these experiments underscore the

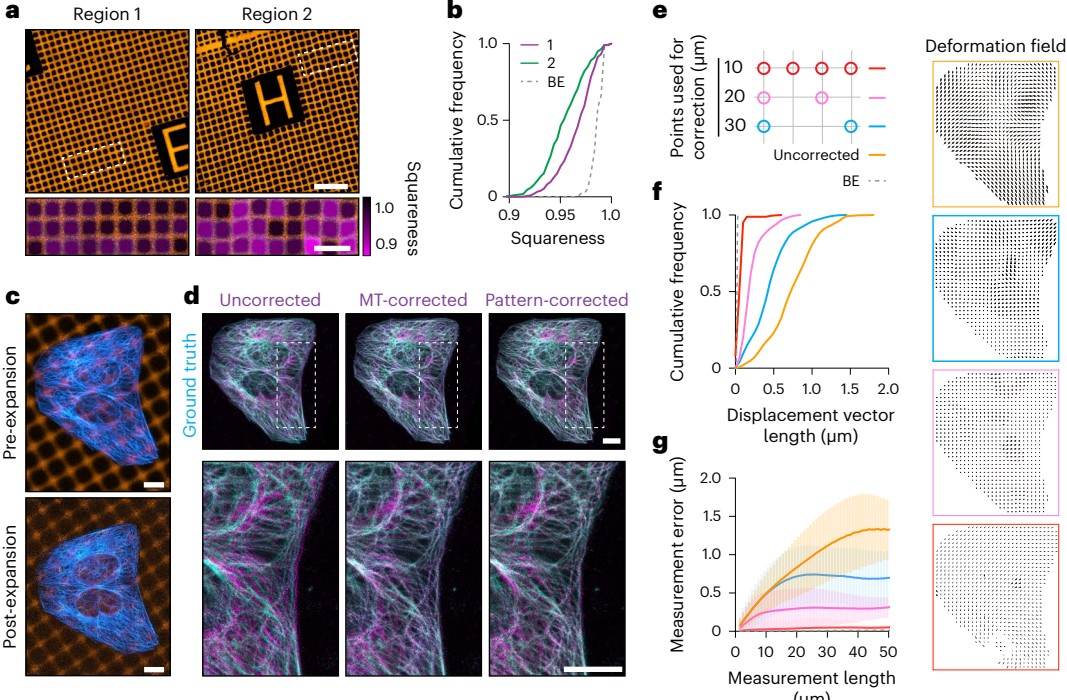

**Fig. 2 | Validation of GelMap-based corrections using pre- and post-expansion imaging. a**, Two representative regions of expanded GelMap grids (20 μm squares) from the same gel. Deformation as defined by squareness is color-coded in magenta in zooms of boxed regions (expansion factor, 9.0 and 8.7 for region 1 and 2, respectively). **b**, Cumulative frequency distribution of squareness values of individual squares in region 1 ($n$ = 453, magenta) and region 2 ($n$ = 396, magenta) from **a**. Dashed line indicates baseline error (BE) resulting from uncertainties in the detection of crossings (Methods). **c**, Pre- and post-expansion images of U2OS cells cultured on nanobody-patterned GelMap grids (grid size, 10 μm; expansion factor, 9.1), stained for tubulin (cyan) and myc-tag (orange). **d**, Linear (left column; uncorrected) and nonlinear transformation (middle column; MT-corrected) of the post-expansion tubulin channel registered to the pre-expanded ground-truth tubulin channel, overlayed with pre-expanded ground-truth image. Pattern-corrected tubulin channel (10-μm grid spacing) overlayed with the pre-expanded ground-truth image (right column, pattern-corrected). Boxed regions correspond to zooms below. Biological replicate can be found in Extended Data Fig. 6e. **e**, Schematic of landmark spacing used for pattern correction. The 10-μm GelMap grid was used for landmark registration, with landmarks removed as indicated to mimic GelMap grids with larger spacing. **f**, Quantification of displacement vector length for equidistantly spaced points (1 μm) in the cellular area for pattern-corrected images registered to the ground-truth tubulin channel. Dashed line indicates BE due to manual landmark selection. Corresponding deformation fields are color-coded and shown on the right. **g**, Quantification of measurement error (deviation from expected distance assuming isotropic expansion) between pairs of points for a measurement length (mean ± s.d.). Dashed line indicates BE due to manual landmark selection. Scale bars, 100 μm (**a**), zoomed regions 40 μm; 10 μm (**c,d**). Scale bars in expanded samples reflect pre-expansion sizes.

need for local quality control mechanisms to standardize quantitative measurements in ExM.

We next explored the effectiveness of GelMap for correcting deformations on the cellular scale. We cultured cells on an NBD-patterned coverslip, fixed and stained for tubulin, amplified the pattern using the orthogonal myc-tag and then examined the corrective power for different grid sizes compared to correction using the pre-expansion image of the tubulin (MT) channel (Fig. 2c,d). First, we visualized the overall deformation by registering and overlaying the pre- and post-expansion tubulin channels using a landmark-based linear similarity transformation (Fig. 2d; uncorrected), which was subsequently corrected using nonlinear thin-plate spline transformation (Fig. 2d; MT-corrected). Next, we used GelMap and first registered the pre-and post-expansion grid images using nonlinear transformation, which was then used to correct the expanded tubulin channel (Fig. 2d; pattern-corrected).

To quantify the corrective power of GelMap, we first determined the baseline error that is introduced during manual landmark selection by rescaling the pre-expansion tubulin channel to the post-expansion dimensions and then registering it to the original image (Extended Data Fig. 6d). Next, we generated three pattern-corrected images, for which we varied the number of landmarks used for registration to represent grid dimensions of 10 × 10 μm, 20 × 20 μm and 30 × 30 μm (Fig. 2e). To determine the residual errors in these GelMap-corrected

images, we then registered these images to the pre-expansion tubulin image. By comparing linear and nonlinear transformations, we could then calculate for each used grid dimension a displacement vector map that shows the remaining deformation for equidistantly spaced points (1 μm) within the cellular area[22]. Inspection of the distribution of vector lengths revealed that finer grids resulted in less residual errors (Fig. 2f and Extended Data Fig. 6f–h). The average residual error decreased from 0.76 ± 0.3 μm (mean ± s.d.) without correction to 0.07 ± 0.05 μm (mean ± s.d.) for correction every 10 μm (baseline error, 0.02 ± 0.005 μm (mean ± s.d.)). As expected, squareness analysis showed that the decrease in displacement vector length following correction corresponded to an improvement in the squareness measurement (Extended Data Fig. 6e).

An alternative measure for deformation is to compare measurement lengths between pairs of points after expansion to the expected distance in the case of uniform expansion and plot the average fractional deviation as a function of the measurement length[1] (Fig. 2g and Extended Data Fig. 6i). Again, we observed that correction using grids with smaller spacing resulted in smaller errors for a given measurement length. While the uncorrected curve saturates at a measurement length of ~45 μm, the curves for measurements after correction saturate around 25 μm, 15 μm and 7 μm for correction using landmarks spaced 30 μm, 20 μm and 10 μm, respectively.

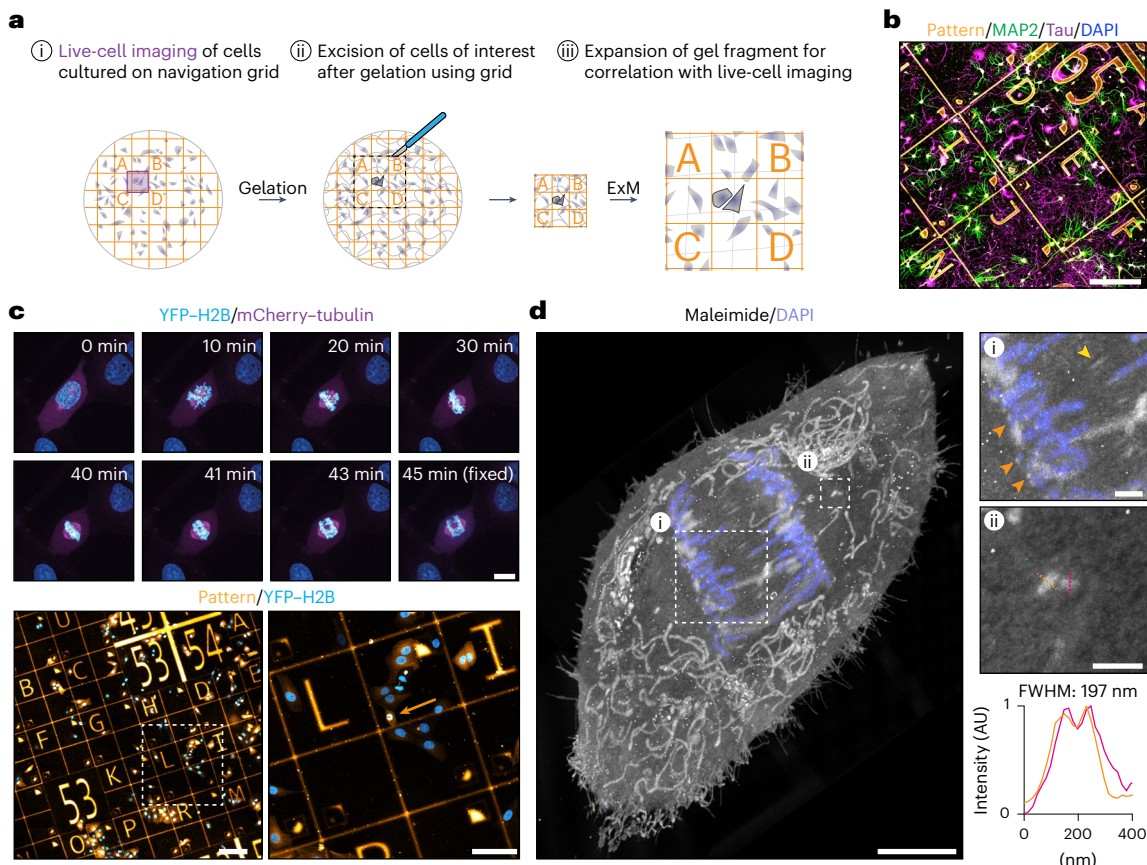

**Fig. 3 | GelMap facilitates sample navigation for correlative live and expansion microscopy. a**, Schematic of the workflow for correlative live and expansion microscopy. (i) Cells of interest cultured on GelMap coverslips are followed live to image a dynamic process. (ii) Cells are incorporated into the ExM hydrogel, digested and the region of interest as indicated by the pattern is excised before expansion. (iii) Gel fragment is expanded and pattern is used to find back the cell of interest for correlation. **b**, Dissociated neurons cultured on GelMap navigation grid, stained for MAP2 (green), Tau (magenta), 4,6-diamidino-2-phenylindole (DAPI) (gray) and myc-tag (orange). **c**, Timelapse imaging of a mitotic cell stably expressing YFP–H2B and mCherry–tubulin. During anaphase, the cell was fixed on the microscope stage. The cell was located on navigation grids and expanded using TREx. **d**, Maximum projection of expanded cell from timelapse imaging in **c** stained for total protein using maleimide and DAPI (expansion factor, 10.7). Zooms are indicated by boxed regions. General protein stain reveals several ultrastructural features, including cell morphology, intracellular organelles such as mitochondria, the spindle midzone (yellow arrow), presumptive kinetochores (zoom 1, orange arrows) and centrioles (zoom 2). Linescan and quantification of centriole width indicated in orange and magenta. FWHM, average full width at half maximum for both linescans; AU, arbitrary units. Scale bars, 200 μm (**b**); timelapse 10 μm (**c**), below, 200 μm (left) and 100 μm (right); 5 μm (**d**), zoomed regions 1 μm. Scale bars in expanded samples reflect pre-expansion sizes.

These experiments have two important implications. First, while all grid sizes can correct for deformation, the absolute corrective power of GelMap is inversely correlated with the grid feature size. Second, as GelMap is effective in correcting deformations via registration to the pre-expanded dimensions, registration to a virtual reference grid of identical dimensions must be equally effective. This in turn negates the need for pre-expansion image acquisition. Therefore, by intrinsically calibrating the ExM hydrogel with a fluorescent grid that scales with the expansion factor and deforms with anisotropy, we have developed a robust quality control method for ExM.

### GelMap facilitates correlative live-ExM experiments

Following up live-cell microscopy experiments with expansion microscopy is a promising approach that can provide new insights into the nanoscale structures that underly cellular dynamics; however, the sample processing steps required after live-cell imaging makes it challenging to locate specific imaged cells in the expanded gels. This is particularly the case for high-expanding gel recipes, as the volume that contains the cell of interest increases with the expansion factor cubed. To address this challenge, we added a coordinate system into our GelMap grids. This would enable us to image a cell, locate the position

of the cell on the grid (Fig. 3a(i)), fix and stain the cells for the structure of interest, gelate the sample and excise the gel region that contains the cell (Fig. 3a(ii)). The excised gel fragment could then be expanded, after which the grid facilitates correlation of the expanded cell with the live-cell acquisition, as well as analysis of local deformations and expansion factor (Fig. 3a(iii)).

We employed various navigation grids (Extended Data Fig. 1b) and tested compatibility of GelMap with a top-coating of substrate to facilitate live imaging of sensitive cells such as neurons, confirming neuronal growth and morphology remained unperturbed (Fig. 3b). To demonstrate correlative expansion microscopy, we cultured cells stably expressing YFP–H2B and mCherry–tubulin on the navigation grid and synchronized the population using thymidine. We picked a cell of interest and followed its progression through mitosis. During anaphase, the cell was rapidly fixed on the microscope stage and located on the navigation grid (Fig. 3c and Supplementary Video 1). We next expanded the coverslip using TREx, excised the region of interest (relying on navigation provided by GelMap) and stained for total protein using maleimide. In the expanded sample, we could use the ultrastructural context provided by maleimide to visualize the cell morphology, organelle distribution and key components of the

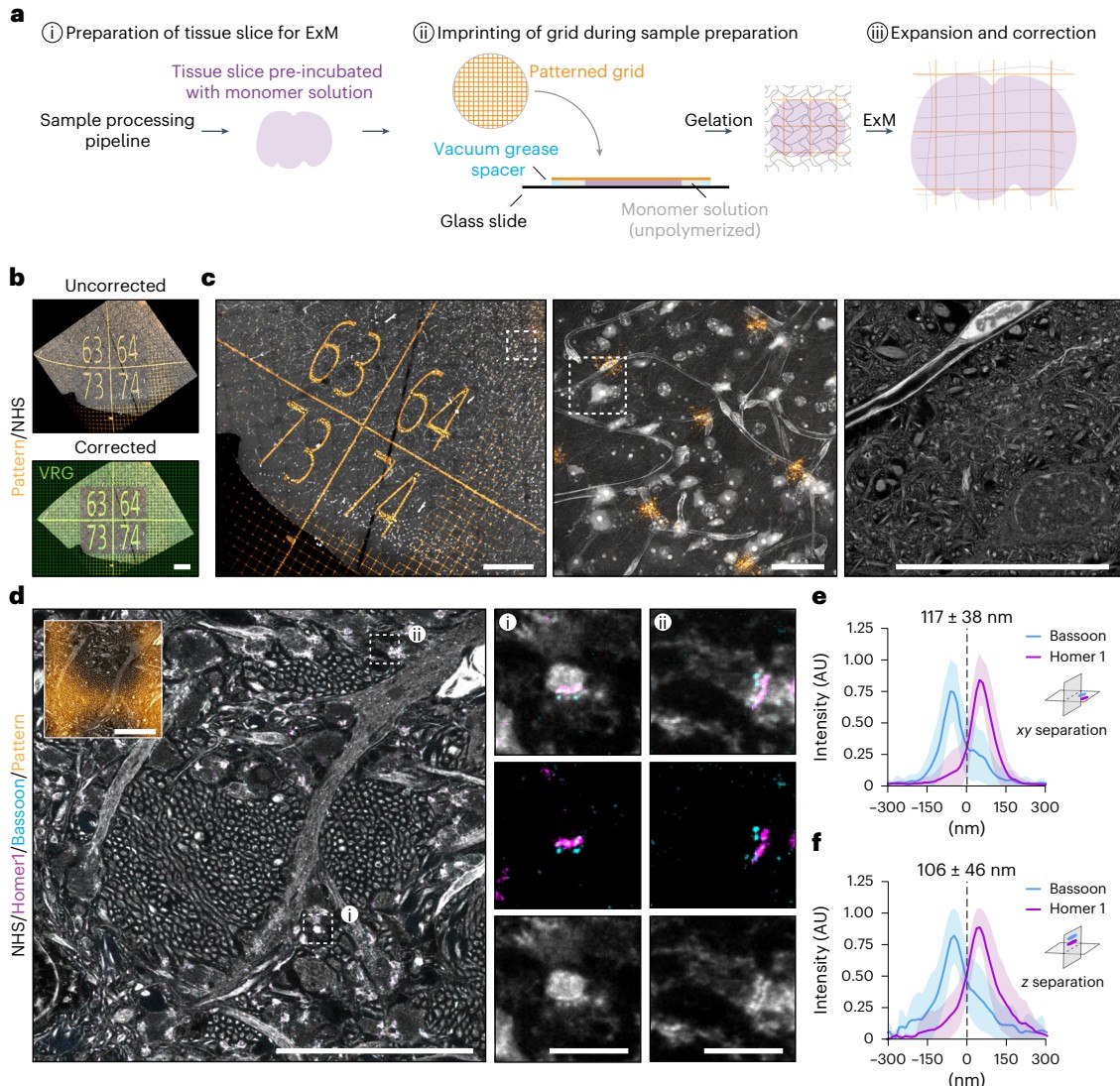

**Fig. 4 | GelMap can be imprinted onto tissue slices and used for correction.**
**a**, Schematic of workflow when using GelMap for tissue slices during sample preparation. **b**, Region of dissected mouse brain tissue (cortex) expanded using TREx (expansion factor, 7.6) and stained for total protein (gray) and myc-tag (orange) and imaged by confocal microscopy (top). The pattern channel was registered to a virtual reference grid (green) to correct for deformation (bottom). The resulting nonlinear transformed protein channel was used for subsequent visualization of corrected tissue organization. **c**, Using GelMap to bridge scales of tissue vasculature organization from $1.3 \times 1.2$ mm (left) to $110 \times 110$ μm (middle) and $25 \times 25$ μm (right) (corrected dimensions) high-resolution zooms of blood vessel with surrounding tissue. Zoomed regions indicated with white boxes. Left and middle panels are maximum projections of 60 frames (corrected z-spacing: 700 nm). **d**, Region of dissected mouse brain tissue (hippocampus) expanded with TREx and stained for Bassoon (cyan), Homer1 (magenta) and total protein (gray) after expansion. Total protein stain reveals dense bundles of thin,

unmyelinated axons (Mossy fibers) projecting perpendicularly to the imaging plane and intersecting with dendrites from neurons in the CA3 region (left). At such intersections, synapses can be identified by the presence of closely apposed protein densities (gray), colocalizing with pre- and postsynaptic scaffold proteins Bassoon (cyan) and Homer1 (magenta), respectively. White boxes mark regions used for the zooms. Corrected GelMap pattern indicated in smaller insert. **e**, Quantification of Bassoon/Homer1 separation for synapses whose synaptic axes are within the imaging plane. Average separation $117 \pm 38$ nm (mean + s.d.)), $n = 69$, three biological replicates. **f**, Quantification of Bassoon/Homer1 separation for synapses whose synaptic axes are approximately perpendicular to the imaging plane. Average separation $106 \pm 46$ nm (mean + s.d.)), $n = 50$, three biological replicates. The difference between the values in **e** and **f** are not statistically significant (NS; Mann–Whitney $U$-test, two-tailed test). Scale bars, 200 μm (**b**); 200 μm (left), 20 μm (middle and right) (**c**); 10 μm, 1 μm (zooms) (**d**). Scale bars in expanded samples reflect pre-expansion sizes.

mitotic spindle, including the spindle midzone, the chromosomes and presumptive kinetochores and both centrioles on either side of the spindle (Fig. 3d, Extended Data Fig. 7b and Supplementary Video 1). GelMap-based determination of the expansion factor (Extended Data Fig. 7a) revealed a centriole diameter of 197 nm, consistent with previously reported values using ExM[5].

### GelMap can be imprinted onto tissue slices

One of the key advantages of ExM is that it can be used to directly expand tissues and organisms[1,6–9]. We therefore tested whether GelMap could

be applied to non-adherent biological samples by incorporating the grid at the stage of sample preparation. Gelation is often performed in a gelation chamber that is closed off during polymerization. We therefore set out to expand mouse brain tissue slices and used a patterned coverslip to close off the gelation chamber before gelation, ensuring contact between the tissue slice and the patterned coverslip (Fig. 4a). Indeed, following expansion and staining of the tissue with an $N$-hydroxysuccinimide (NHS) general protein stain, we visualized the imprinted fluorescent pattern in combination with the tissue (Fig. 4b). This revealed extensive distortions of the grid that indicate

local deformations. These distortions could be subsequently corrected by aligning the expanded image to a virtual reference grid followed by nonlinear transformation (Fig. 4b and Supplementary Video 2).

Using GelMap, we could bridge multiple layers of tissue organization; from a low-resolution overview tilescan ($1.3 \times 1.2$ mm, corrected dimensions), to a higher resolution overview of the local arrangement of cells and tissue vasculature ($110 \times 110$ µm, corrected dimensions), down to high-resolution ultrastructural details of neuronal somata, surrounding neuropil and the vascular endothelium ($25 \times 25$ µm, corrected dimensions) (Fig. 4c and Supplementary Video 2). Next, we expanded mouse brain tissue and stained for total protein in combination with post-expansion labeling of pre- and postsynaptic marker proteins Bassoon and Homer1 using TREx (Fig. 4d). Putative synapses could be readily observed based on the presence of closely apposed pre- and postsynaptic protein densities that were positive for Bassoon and Homer1, respectively (Fig. 4d; zooms). After correction for the expansion factor using GelMap, we quantified the mean separation between Bassoon and Homer1 for synapses where both the pre- and post-synapse were in the imaging plane and found an average separation of $117 \pm 38$ nm (mean ± s.d.) (Fig. 4e), which is consistent with previously reported values between 90 and 150 nm using ExM[1,14,23–25], as well as non-ExM modalities[26–29]. We then also quantified synapse separation for synapses oriented toward the z axis and found that the average separation of $106 \pm 46$ nm (mean ± s.d.) was not significantly different (Fig. 4f). This indicates that the GelMap-based measurement for xy expansion is a good estimation for expansion along the z axis.

These results demonstrate that the GelMap approach can be added to various existing ExM sample processing pipelines to provide intrinsic calibration and deformation mapping for expanded tissue.

## Discussion

Here, we introduce GelMap as an approach to intrinsically calibrate ExM hydrogels through the incorporation of a fluorescent grid that scales with the expansion factor and deforms upon anisotropic expansion. Our approach overcomes three key challenges of ExM. First, it provides a robust way to determine the exact expansion factor in relevant regions of interest. Second, it enables one to routinely quantify and correct local deformations without the need for a pre-expanded reference image, which is critical for expansion variants where pre-expansion imaging is not possible (for example when using post-expansion labeling). Third, it encodes a fluorescent coordinate system to the hydrogel that aids sample navigation of expanded samples. Notably, GelMap can be easily incorporated into existing ExM workflows to provide a standardized quality control step.

In this work, we used deep-UV-based micropatterning to create GelMap grids with grid sizes down to 10 µm and demonstrate that this provides sufficient resolution for deformation mapping and correction at the subcellular level. For even finer corrections, alternative patterning approaches, such as microcontact printing (µCP) may be considered[30]. Furthermore, the current implementation of GelMap requires antibody-based amplification of the protein used for patterning, we anticipate that signal intensities after expansion can still be optimized by engineering additional labeling sites into the nanobody or by increasing the anchoring to the hydrogel, for example through introduction of an expansion cassette[31]. Although such amplification can hinder the use of primary antibodies of the same host species for specific labeling due to signal overlap, in practice we find that this problem is minimal for structures that are positioned away from the coverslip surface (Fig. 4d; both Bassoon and Myc labeled with primary antibodies raised in rabbit and amplified with the same secondary antibody).

Currently, the most straightforward implementation of Gel-Map is the use of pre-patterned substrate-attached proteins for creating two-dimensional grids. While we demonstrated that the two-dimensional GelMap grids provide a good estimate for the expansion along the z axis, they cannot be directly used to correct

deformations that may occur along this axis. To overcome this limitation, we envision the use of coverslip-based GelMap grids on both sides of the gel, followed by deformation mapping and side-to-side interpolation may provide a good proxy for any shearing and other deformation within the gel. Alternatively, we have also shown that grids can be created by structured photoconversion of gel-incorporated fluorophores before expansion; however, using photoactivatable rhodamine we experienced noticeable signal loss, which suggest that photoactivatable approaches that allow for signal amplification might be better suited, for example using photocleavable DNA in combination with a hybridization chain-reaction[32]. When used in combination with multi-photon conversion to ensure localized conversion in the third dimension, this may enable creating true 3D grids.

In summary, GelMap provides a simple and robust method to calibrate expanded gels, to map and correct expansion-induced deformations and to facilitate sample navigation in correlative microscopy approaches. We anticipate that this will aid the development and validation of new gelation and expansion protocols and facilitate the widespread implementation of ExM into imaging applications that require robust and reliable calibration, both in basic science and in clinical applications.

## Online content

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

# Methods

## Photomask design

The CleWin 5 (WieWeb) layout editor software was used for designing chrome–quartz photomasks. The following base design was used: a 2 × 2 cm area consisting of 100 (00–99) 2 × 2 mm tiles, containing five 400 × 400 µm numbers (Extended Data Fig. 1a). Variants on the base design follow either a repeated grid pattern for use in mapping gel deformation, or a coordinate letter system to further aid in sample navigation (Extended Data Fig. 1b). Photomasks (1× master, 5′ × 5′ × 0.090′, grade B) were fabricated by Toppan (Toppan Photomasks), with features (numbers, lines) chrome and blank spaces quartz.

## Conjugation

Conjugation of protein or nanobody was performed with an eightfold (laminin; Roche, 11243217001) or fourfold (R2-myc-his; a gift from S. Oliveira) molar excess of NHS ester (ATTO 425, ATTO 647N; ATTO-TEC, Alexa 488, Alexa 594; Sigma-Aldrich) in a minimal volume (<50 µl) of Milli-Q water (MQ). Protein/nanobody and NHS ester were mixed by vortexing and incubated overnight at 4 °C before storing 10-µg aliquots at −20 °C. Fibrinogen pre-conjugated with Alexa Fluor 647 (Invitrogen, F35200) was reconstituted in 0.1 M sodium bicarbonate buffer, pH 8.3, and aliquots were stored at −20 °C.

## Preparation of patterned coverslips

Coverslips (Ø18 mm, no. 1.5; Marienfeld, 107032) were washed for 10 min in acetone, sonicated for 20 min in 50% methanol/MQ and sonicated for a further 20 min in 0.5 M potassium hydroxide before washing three times in MQ. Coverslips were then washed in 100% ethanol and dried under a flow of nitrogen before storage. Cleaned coverslips were activated with air plasma (PDC-002, Harrick Plasma) for 5 min before incubation with 2.5 µg fluorescent protein in 100 µl MQ per coverslip for 1 h at room temperature. Coverslips were dried at room temperature on a clean tissue for 30 min before being exposed to deep-UV (~250 nm) through a clean chrome/quartz photomask for 4 min, using a UVO cleaner (Jelight Company, 42–220). Patterned coverslips were washed with PBS for storage at 4 °C. Photomasks were cleaned by 10 min acetone and isopropanol alcohol washes, before rinsing in MQ, washing with 100% ethanol and drying under a flow of nitrogen. A polyvinyl alcohol (PVA) sponge (Ramer) was used to wipe the photomask during washes. Photomasks were exposed to air plasma for 20 min before patterning for further cleaning.

Before cell seeding, patterned coverslips were sterilized with 70% ethanol and washed three times in sterile PBS. For neuronal experiments, patterned coverslips were top-coated with 37.5 µg ml$^{-1}$ poly-L-lysine (Sigma-Aldrich, P8920) and 1.25 µg ml$^{-1}$ laminin (Roche, 11243217001) in 0.1 M borate buffer, pH 8.5. Laminin-patterned coverslips were therefore not used for neuronal experiments to avoid unspecific labeling of the top-coat during GelMap grid amplification. We provide a graphical workflow for preparation of GelMap coverslips in Extended Data Fig. 2.

## Photo-uncageable rhodamine patterning within polymerized hydrogels

To generate a photoactivatable molecule that could be covalently incorporated into polymerized hydrogels, a two-step conjugation was performed of NVOC2-Q-rhodamine-5-PEG3-azide (Sigma-Aldrich 768693) to acryloyl-X SE (AcX) (Thermo Fisher, A20770) using a sulfo-DBCO-amine linker (Broadpharm, BP-23309). First, Sulfo-DBCO-amine (40 mM) was reacted with AcX (21 mM) in dimethyl-sulfoxide (DMSO) for 1 h at room temperature (RT) to form DBCO-AcX. Next, a 2-4× molar excess of NVOC2-Q-rhodamine-5-PEG3-azide was reacted with DBCO-AcX for 1 h at RT. By mixing the final formed molecule (NVOC2-Q-rhodamine-5-PEG3-AcX) into the unpolymerized gel solution at a final concentration of 120 µM, the dye becomes covalently linked to the ExM hydrogel during gelation.

## Cell culture and cell synchronization

U2OS (ATCC) cells were cultured in DMEM high glucose medium (Capricorn DMEM-HPSTA) supplemented with 9% fetal bovine serum (Corning, 35_079_CV) and 1% penicillin/streptomycin (GIBCO, 15140-122). Primary hippocampal neurons were maintained in Neurobasal medium (GIBCO, 21103049) supplemented with 1% B27 (GIBCO, 17504001), 0.5 mM L-glutamine (GIBCO, 25030-081), 15.6 µM glutamate (Sigma-Aldrich, G1251) and 1% penicillin/streptomycin. For imaging of mitosis, U2OS cells expressing YFP–H2B and mCherry–tubulin (a gift from J. Raaijmakers and R. Medema, NKI Amsterdam) were synchronized using a double thymidine block[33]. In short, cells were seeded at 25% confluency, 16 h after seeding incubated with 2 mM thymidine (Millipore, 6060) for 24 h, incubated with complete medium for 9 h, blocked again using 2 mM thymidine for 29 h and finally released from second block in complete medium 12 h before imaging.

## Fixation and immunostaining

For correlative deformation mapping experiments and for the comparison across ExM variants, cells were extracted using pre-warmed 0.35% Triton X-100 + 0.2% glutaraldehyde in MRB80 for 1 min, followed by 4% paraformaldehyde fixation in PBS for 10 min. For experiments using protein stains, cells were fixed with pre-warmed (37 °C) 4% paraformaldehyde + 0.1% glutaraldehyde + 4% sucrose in PBS for 10 min. Next, cells were washed with PBS and permeabilized with PBS + 0.2% Triton X-100. Blocking and antibody-labeling steps were performed with 3% bovine serum albumin in PBS. The patterned grids were amplified by immunostaining for laminin, fibrinogen or myc-tag (R2-myc-his) along with antibody labeling for specific structures. The following primary antibodies were used in this work: rabbit anti-laminin (1:100 dilution; Abcam, ab11575), rabbit anti-myc-tag (1:100 dilution; Cell Signaling Technology, 2272), rabbit anti-fibrinogen (1:100 dilution; Abcam, ab34269), rat anti-tubulin YL1/2 (1:200 dilution; Abcam, ab6160), mouse anti-tau (1:500 dilution; Sigma-Aldrich, MAB3420), chicken anti-MAP2 (1:500 dilution; Abcam, ab5392), mouse anti-Bassoon (1:250 dilution; Enzo, SAP7F407) and rabbit anti-Homer1 (1:250 dilution; Synaptic Systems, 160003). The following secondary antibodies were used in this work: goat anti-rabbit IgG Alexa Fluor 594 (Invitrogen, A-11037), goat anti-rabbit IgG Alexa Fluor 488 (Invitrogen, A-11029), goat anti-rat IgG Alexa Fluor 488 (Invitrogen, A-11006), goat anti-chicken IgY DyLight 488 (Invitrogen, SA5-10070) and goat anti-mouse IgG Alexa Fluor 647 (Invitrogen, A-21236), all at 1:250 dilution (pre-expansion labeling) or 1:500 dilution (post-expansion labeling).

## Tissue fixation and sectioning

All animal experiments were carried out according to the regulations of Utrecht University and in agreement with Dutch law (Wet op de Dierproeven, 1996) and European regulations (EC Directive 2010/63/EU). Up until the moment of perfusion, mice were group-housed (2–4 per cage) in a temperature- and humidity-controlled room (22 ± 2 °C and 60–65%, respectively) under a 12 h light–dark cycle (lights on at 7:00) with ad libitum access to water and standard laboratory chow (Special Diet Services (SDS), product code CRM(E)). The 10-month-old, male TRAP2 heterozygous mice (Jax, 030323) were transcardially perfused with ice-cold fixative solution (4% formaldehyde and 20% acrylamide in PBS, pH 7.4). Brains were removed and post-fixed in fixative solution overnight at 4 °C. Fixed brains were washed three times for 1 h in PBS at RT and cut coronally into 100-µm thick sections using a vibratome (Leica VT1000 S). After cutting, fixed brain sections were stored in PBS at 4 °C.

## Tenfold robust expansion microscopy

TREx microscopy was performed as previously published[14]. Cells were treated with 100 µg ml$^{-1}$ acryloyl-X SE (AcX) (Thermo Fisher, A20770) in PBS overnight at RT or 0.8% formaldehyde + 1% acrylamide at 37 °C for 5 h. TREx gelation solution was prepared containing 1.1 M

sodium acrylate (SA), 2.0 M acrylamide (AA) (Sigma-Aldrich, A4058), 50 ppm $N,N'$-methylenebisacrylamide (bis) (Sigma-Aldrich M1533), PBS (1×), 0.15% APS (Sigma-Aldrich, 215589) and 0.15% TEMED (Bio-Rad, 1610800). The 4 M SA stocks were prepared as previously described[14] by neutralizing acrylic acid (Sigma-Aldrich, 147230) with 10 M sodium hydroxide to a final pH of 7.5–8. Stocks were stored long-term at −20 °C. Gels were prepared in a gelation chamber consisting of a parafilm covered glass slide with a silicone gasket (Sigma-Aldrich, GBL66410). Gels were left to gelate for 1 h at 37 °C. For proteolytic homogenization, samples were transferred to a 12-well plate and digested with 7.5 U ml$^{-1}$ Proteinase-K (Thermo Fisher, EO0491) in TAE buffer (containing 40 mM Tris, 20 mM acetic acid and 1 mM EDTA) supplemented with 0.5% Triton X-100, 0.8 M guanidine-HCl and DAPI for 4 h at 37 °C. For correlative experiments, regions of interest were located after digestion using fluorescent grid, excised and expanded using an EVOS imaging system equipped with Plan Fluor ×10/0.3 objective (Thermo Fisher Scientific). The gel was transferred to a Petri dish, water was exchanged 2 × 30 min and the sample was left in MQ to expand overnight. For samples stained for total protein using maleimide, gels were washed 2 × 15 min in PBS after gelation and incubated with 20 µg ml$^{-1}$ Atto 647N maleimide (Atto-Tec, AD 647N) in PBS prepared from a 20 mg ml$^{-1}$ stock solution in DMSO for 1.5 h at RT with shaking. Next, samples were rinsed in PBS, digested and expanded as described above. For non-proteolytic homogenization after gelation, gels were incubated in a denaturation buffer (50 mM Tris (pH 8.8), 5% SDS and 200 mM NaCl) for 15 min at RT followed by 1 h at 95 °C. Gels were either processed for post-expansion labeling or directly expanded by several washes with excess MQ.

## Post-expansion labeling

For post-expansion labeling, gels were cut into smaller pieces and first incubated with primary antibody (1:250 dilution) in PBS containing 0.1% Triton X-100 and 1% BSA for 24 h at 4 °C with gentle shaking. Next, gels were washed 4 × 30 min with PBS + 0.1% Triton X-100 at RT, followed by secondary antibody incubation (1:500 dilution) in PBS + 0.1% Triton X-100 + 1% BSA for 24 h at 4 °C with gentle shaking. Gels were again washed 4 × 30 min with PBS + 0.1% Triton X-100 and afterwards expanded in MQ.

## Expansion of tissue and incorporating patterned grids during sample processing

For expansion of tissue, brain slices were pre-incubated with TREx gelation solution for 30 min on ice in the presence of 15 µg ml$^{-1}$ 4-hydroxy-TEMPO (Sigma-Aldrich, 176141) to delay premature gelation. To create the gelation chamber, tissue slices were laid out on a microscopy slide with four dabs of vacuum grease surrounding the slice. A GelMap coverslip was placed with the protein grid facing the tissue on top of the dabs of vacuum grease and pressed down ensuring contact with the tissue slice. The gelation chamber was filled with gelation solution from the side and transferred to 37 °C for 1 h. After gelation, the gel surrounding the tissue was trimmed and the sample was denatured for 3 h at 80 °C in disruption buffer containing 5% SDS, 200 mM NaCl and 50 mM Tris, pH 7.5. After disruption, gels were washed in PBS and either first processed for post-expansion labeling as described above or directly stained with 30 µg ml$^{-1}$ NHS ester conjugated to ATTO488 (ATTO-TEC) for 2 h at RT. After staining, gels were washed and expanded in MQ.

## Validation using common expansion variants

After fixation, cells were either treated with 100 µg ml$^{-1}$ AcX overnight at RT or with 0.8% formaldehyde + 1% acrylamide at 37 °C for 5 h. For all recipes the gelation chamber was constructed as described above. Pro-ExM was performed as described previously[2]. In short, cells were incubated in gelation solution consisting of 8.625% SA, 2.5% AA, 0.1% bis, 1× PBS, 2 M sodium chloride, 0.2% TEMED and 0.2% APS for 15 min on ice, followed by 1 h polymerization at 37 °C. Proteolytic

homogenization with Proteinase-K and expansion in MQ were performed as described above.

MAP was performed as described in[8]. In short, cells were incubated with gelation solution consisting of 7% SA, 20% AA, 0.1% bis, 1× PBS, 0.5% TEMED and 0.5% APS for 1 min on ice, followed by 1 h polymerization at 37 °C. For homogenization, gels were incubated in denaturation buffer containing 5% SDS, 200 mM NaCl and 50 mM Tris, pH 8.8, for 15 min at RT, followed by 1 h at 95 °C. After several washes with excess MQ and PBS, gels were cut into smaller pieces and processed for post-expansion labeling as described above. Gels were again washed 4 × 30 min with PBS + 0.1% Triton X-100 and afterwards expanded in MQ.

Iterative pan-ExM was performed as described in[18]. In short, cells were embedded in the first gel by incubating in gelation solution containing 19% SA, 10% AA, 0.1% DHEBA (Sigma-Aldrich, 294381, lot 0000189275), 1× PBS, 0.25% TEMED and 0.25% APS for 15 min at RT followed by 1.5 h at 37 °C. Next, gels were homogenized in acidic denaturation buffer (pH 6.8) for 15 min at RT, followed by 1 h at 75 °C. After several washes with excess MQ, gels were cut into approximately 1 × 1 cm squares for re-embedding into a non-expanding, stabilizing gel. Cut gels were incubated three times for 20 min with the second gelation solution consisting of 10% AA, 0.05% DHEBA, 0.05% TEMED and 0.05% APS on ice. Next, gels were placed on a microscope slide and excess gelation solution was tapped off thoroughly with Kim wipes. To form a second gelation chamber, gels were covered with a glass coverslip. The gelation chamber was placed in a humidified container, filled with nitrogen, sealed and polymerized for 1.5 h at 37 °C. After polymerization of the stabilizing gel, gels were incubated three times for 15 min on ice in a third gelation solution consisting of 19% SA, 10% AA, 0.1% bis, 1× PBS, 0.05% TEMED and 0.05% APS. Gels were transferred to humified, nitrogen-filled container as described above and incubated for 1.5 h at 37 °C for polymerization. After polymerization of the third gel, the DHEBA crosslinker was cleaved by incubating gels for 1 h in 0.1 M NaOH at RT. Next, gels were washed multiple times for 30 min in PBS until the pH of the PBS stabilized to pH 7.4. Post-expansion labeling was performed as described above.

## Imaging acquisition

Expanded gels were trimmed using a scalpel blade to fit into an Atto-fluor Cell Chamber (Molecular Probes, A-7816), or into a custom designed 3D printed imaging chamber. ExM and pre-expansion images were acquired using a Leica TCS SP8 STED 3X microscope equipped with HC PL APO ×20/0.75 dry and HC PL APO ×86/1.20 W motCORR STED (Leica, 15506333) water objectives. A pulsed white laser (80 MHz) and a 405 nm DMOD Flexible UV laser were used for excitation. The internal Leica GaAsP HyD hybrid detectors were used with a time gate of 1–6 ns. The setup was controlled using LAS X. For photo-uncaging of rhodamine to create patterns in polymerized hydrogel, 50 × 50-µm field of views were scanned with 405 nm laser light to create a boxed pattern.

For imaging of pre-expanded neurons cultured on patterned coverslips, a Zeiss LSM 700 confocal setup consisting of an AxioObserver Z1 microscope with a Plan-Apochromat 20×/0.8 dry objective was used. The setup was controlled using ZEN.

For live-cell imaging of mitosis, coverslips were mounted in complete medium and images acquired using a ×60 (Plan Apo VC, NA 1.4; Nikon) oil-immersion objective on a Spinning Disc (Yokogawa CSU-X1-A1) Nikon Eclipse Ti microscope with Perfect Focus System equipped with a sample incubator (Tokai-Hit) and an Evolve 512 EMCCD camera (Photometrics), controlled with MetaMorph v.7.7 software (Molecular Devices). Cobolt Calypso 491 nm and Cobolt Jive 561 nm lasers were used for excitation. Images were acquired every 60 s and fixed on the stage during imaging with pre-warmed fixative.

For imaging the chrome–quartz photomask and validating antibody amplification, an AMG EVOS digital inverted microscope was used, equipped with AMG ×4/0.13 Plan LWD PH and AMG ×10/0.3 Plan FL objectives.

## Image analysis

All imaging processing was performed using FIJI v.1.54e[34]. For nonlinear transformation of post-expanded images using landmark-based registration of expanded samples to a (virtual) reference grid the plugin BigWarp[35] was used. We provide a graphical workflow for correction using BigWarp in Extended Data Fig. 3. All scale bars in figures represent pre-expansion dimensions, following correction for expansion factor using GelMap. Estimation of macroscopic expansion factor by unbiased participants was obtained by measuring the size of the expanded gel with a ruler and comparing it to the size of the original silicone mold.

For deformation correction using GelMap, regular landmarks provided by the grid were registered to either the pre-expanded grid or a virtual reference grid followed by nonlinear thin-plate spline transformation. For Fig. 2e–g and Extended Data Fig. 6g–i, the total deformation was determined by landmark-based registration of pre- and post-expansion images of the microtubule cytoskeleton. The pattern correction was performed as described above and the residual error was quantified by registering the corrected image of the MT cytoskeleton to the pre-expansion ground truth. By comparing linear similarity transformation to nonlinear thin-plate spline transformation, the absolute deformation for equidistantly spaced points (1 μm) within the cellular area was calculated[22]. Comparing measurement lengths between pairs of points after expansion to the expected distance in the case of uniform expansion and plotting of the average fractional deviation as a function of the measurement length was performed as previously described[14]. In all corrected post-expansion images, the expansion factor was determined from the BigWarp transformation[22].

Expansion anisotropy was quantified using a semi-automatic 'squareness' measurement of deformation. This measure was defined by the detection of the corners of each square in the GelMap grid, followed by determination of the length of the square diagonals as well as the intersection angles of these diagonals. Squareness was then determined as the average of two metrics: (1) the ratio between the shorter and the longer diagonal length and (2) the ratio between the smaller and the larger intersection angles of the diagonals. This metric takes into account both stretching and skewness and will yield a value between 0 and 1, with 1 corresponding to a perfect square and 0 corresponding to a completely flattened square, where two of the opposite corners are overlapping. The expansion factor was calculated for each square by dividing the measured area by the known pre-expansion area.

BEs for both deformation correction and squareness measurements were determined by rescaling the pre-expansion images to ensure comparable pixel sizes, based on measuring the average full width at half maximum of GelMap grid intersections in post-expansion images (as in Extended Data Fig. 6d). These resampled images were then registered onto the original image or used for squareness measurements to estimate the baseline error due to any variability in landmark selection.

For the synapse separation quantification, images were first corrected for expansion and then resampled to the same pixel size per image. Linescans were manually drawn over visibly separated Bassoon–Homer1 pairs, perpendicular to the synaptic cleft using FIJI. Next, linescans for Bassoon and Homer1 were normalized, the peak-to-peak distance determined, aligned to the midpoint of the peak-to-peak distance and averaged using GraphPad Prism 9. Data visualization and statistics were performed using GraphPad Prism 9.

## Reporting summary

Further information on research design is available in the Nature Portfolio Reporting Summary linked to this article.

## Data availability

The datasets generated and/or analyzed during the current study are available on Figshare[36] (https://doi.org/10.6084/m9.figshare.21923418). The design files used to generate photomasks for the current study are available upon request.

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

## Acknowledgements

We thank W. Nijenhuis for helpful advice and E. Katrukha for helpful discussions about image analysis. We thank J. Raaijmakers and R. Medema, NKI Amsterdam for U2OS cells expressing YFP–H2B and mCherry–tubulin. We thank S. Oliveira and A. Di Maggio for the R2-myc-His nanobody. This work was supported by EMBO through a long-term fellowship (EMBO ALTF 407-2017 to M.B.), by the European Research Council (ERC Consolidator grant 819219 to L.C.K.) and by the Eindhoven, Wageningen Utrecht Alliance through the Centre for Living Technologies.

## Author contributions

H.G.J.D., J.B.P. and L.C.K. designed the study. H.G.J.D., J.B.P. and A.S. created reagents and performed experiments. M.B. established protein micropatterning in our laboratory. A.S., I.K. and F.J.M. contributed tissue experiments. H.G.J.D. and J.B.P. analyzed the data and prepared figures. H.G.J.D., J.B.P. and L.C.K. wrote the manuscript with input from all authors. A.A. provided advice and guidance. L.C.K. supervised the study.

## Competing interests

H.G.J.D., J.B.P. and L.C.K. have filed a patent application covering the presented methods. The remaining authors declare no competing interests.

## Additional information

**Extended data** is available for this paper at https://doi.org/10.1038/s41592-023-02001-y.

**Correspondence and requests for materials** should be addressed to Lukas C. Kapitein.

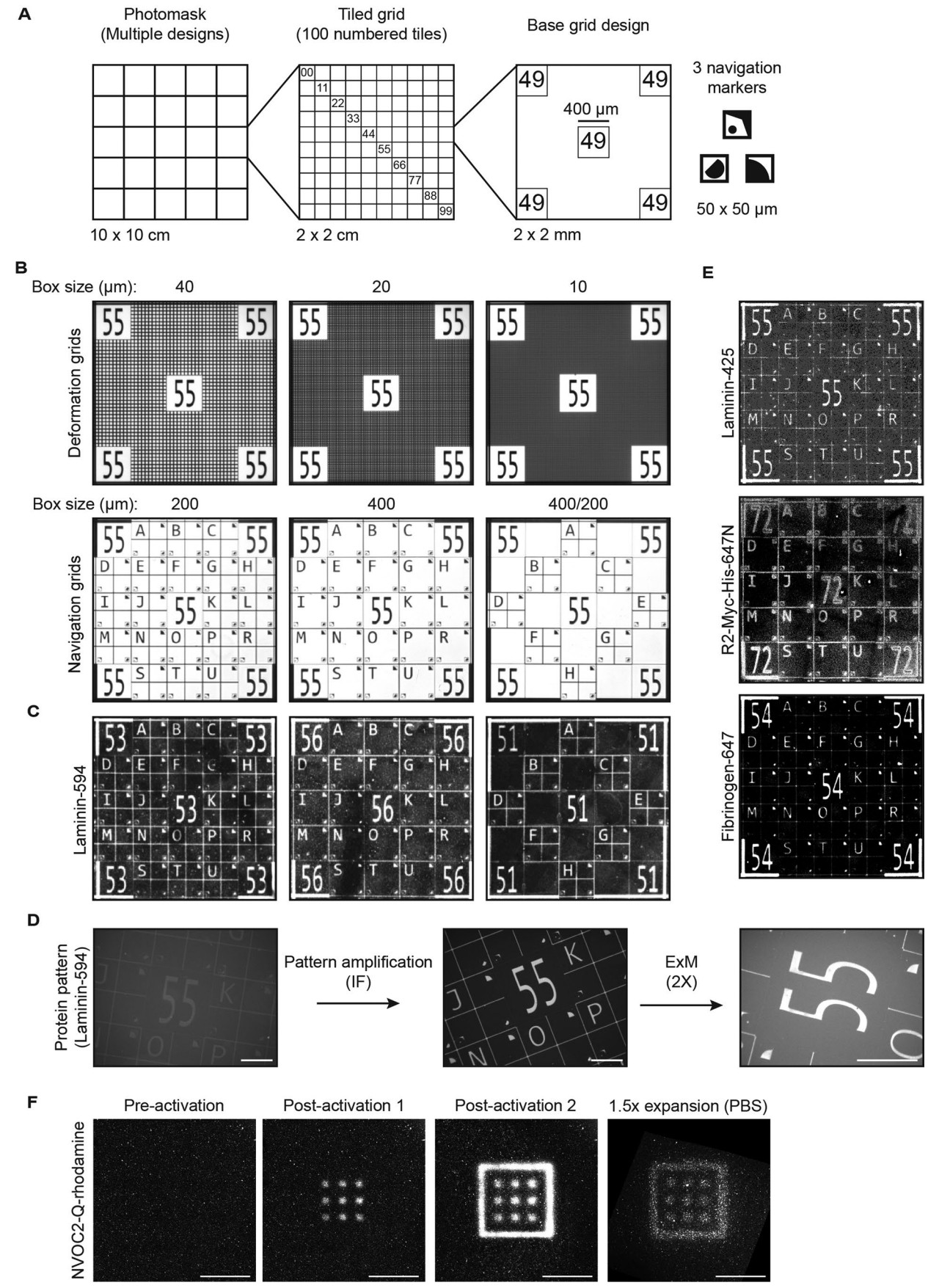

**Extended Data Fig. 1 | See next page for caption.**

**Extended Data Fig. 1 | GelMap pattern design and validation. a**) Photomask design showing tiled grid and the base grid design that can be adapted depending on the experiment. **b**) Widefield images of the photomask containing various grid designs focused on deformation mapping (top) and navigation (bottom). **c**) Laminin conjugated to Alexa 594 was patterned onto a glass coverslip using photolithography using various photomask designs. **d**) Widefield images of a laminin-594 patterned coverslip after patterning (left), after amplification of the signal using antibody labeling against laminin (middle) and moderately expanding using TREx (right). **e**) GelMap coverslips can be produced using multiple proteins (laminin, nanobody (R2-Myc-His) and fibrinogen) and various conjugated dyes. **f**) Alternative implementation of GelMap: an acrylate-modified photo-uncageable rhodamine was incorporated during gelation. Using targeted illumination, a fluorescent pattern was generated into the polymerized hydrogel. Finally, the pattern remains stable in the hydrogel and could be moderately expanded (expansion factor: 1.5) and aligned to pre-expanded dimensions for correction. Scale bars: base grid design: 2 × 2 mm, D: 200 μm, F: 20 μm. Scale bars in expanded samples reflect pre-expansion sizes.

**1. Prepare coverslips**

Coverslips → acetone wash → Successive methanol/KOH washes with sonication in water bath → MQ/ethanol washes → dry under N₂ flow → Plasma clean for 5 min

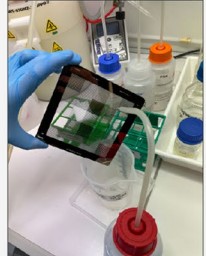 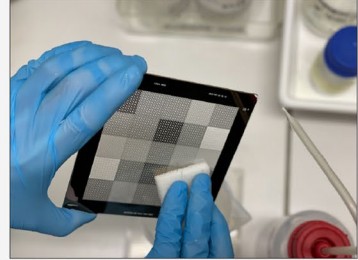 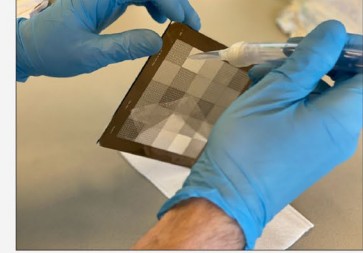 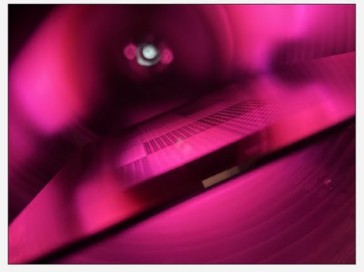

**2. Clean photomask before patterning**

Successive acetone/isopropanol/MQ/ethanol washes → Wipe with polyvinyl alcohol sponge → dry under N₂ flow → Plasma clean for 20 min

**3. Uniformly coat coverslips using (fluorescently labeled) protein**

Incubate coverslips on drop of protein for 1h → Dab off excess fluid and air dry

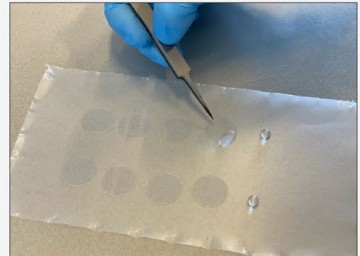 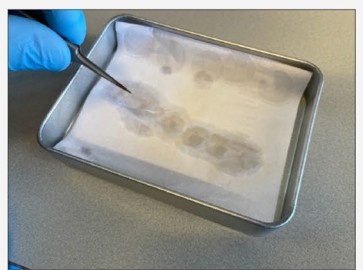

**4. Position coverslips under photomask**

Align coverslip with desired pattern using vacuum holder → Close holder off using photomask and draw vacuum

**5. Pattern coverslips using deep-UV exposure**

While maintaining vacuum, place holder on tray → Expose to deep-UV for 4 min

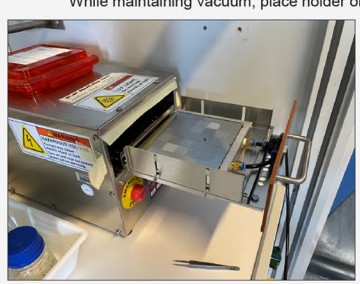 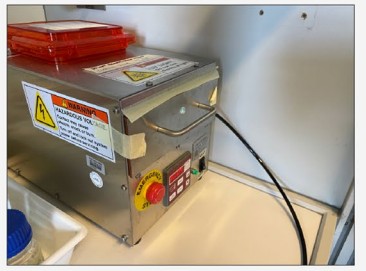

**6. Collect coverslips and validate patterning**

Release vacuum and transfer to well plate → Check pattern

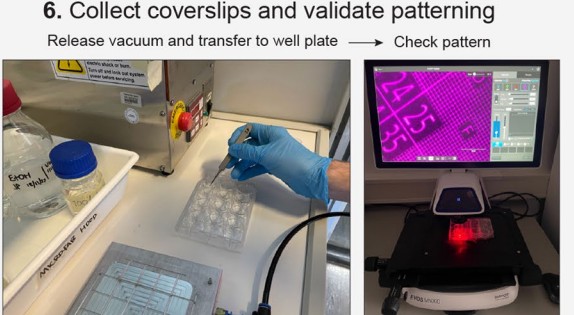

**Extended Data Fig. 2 | Graphical workflow for the production of GelMap coverslips. 1)** Coverslips are thoroughly cleaned by first washing in acetone, and then successive methanol/potassium hydroxide washes with sonication. After rinsing in MQ and ethanol, coverslips are dried, and plasma cleaned. **2)** Photomasks are thoroughly cleaned by first washing in acetone, then isopropanol alcohol, while wiping with a PVA sponge. After rinsing in MQ and ethanol, photomasks are dried, and plasma cleaned. **3)** Cleaned coverslips are coated by incubating on a drop of pre-conjugated fluorescent protein for 1 hour, before air drying for long-term storage at room temperature. **4)** Coated coverslips are placed on a vacuum holder to ensure positioning of the photomask directly above the coverslips. **5)** The coverslips are exposed to deep-UV (~250 nm) through the photomask for 4 mins. **6)** Patterned coverslips can be stored dry at room temperature, and the patterning efficiency can be immediately checked with a fluorescent microscope.

**1.** Open pre- and post-expansion images
(or use virtual reference grid (VRG))

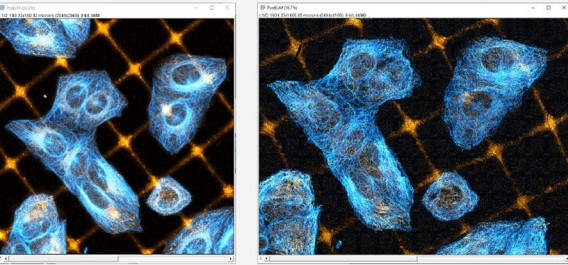

**2.** Open Big Warp and select postExM image as
moving image and preExM/VRG as target image

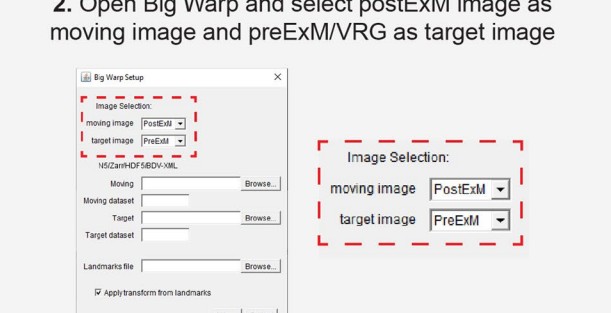

**3.** Manually align views and adjust
LUTs as necessary

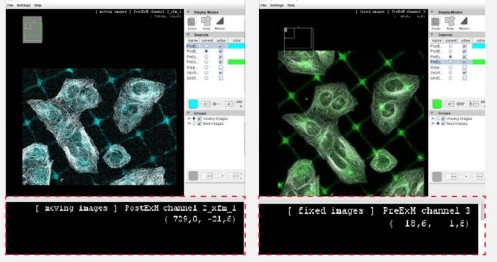

**4.** Select 4 landmarks for initial alignment

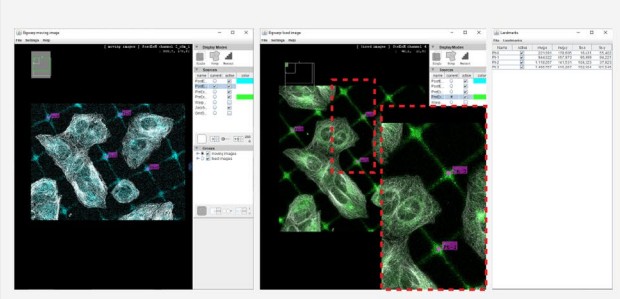

**5.** Transform moving image
(default setting: non-linear Thin Plate Spline (TPS))

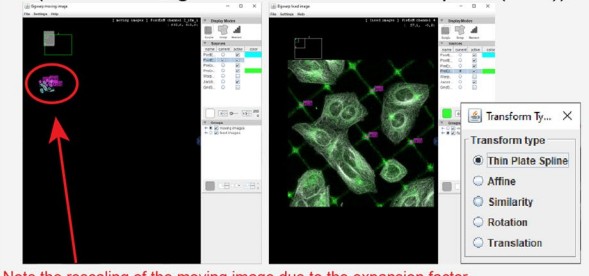

Note the rescaling of the moving image due to the expansion factor

**6.** Adjust viewer to refine using
additional landmarks

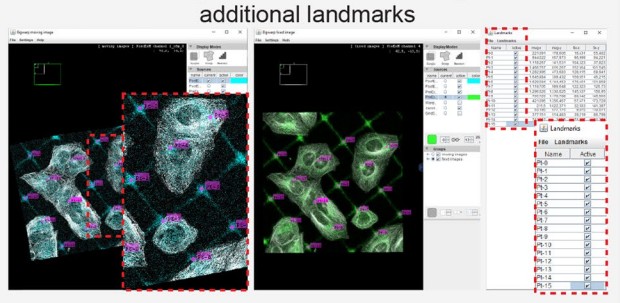

**7.** Overlay moving and target to check alignment
and toggle between similarity (linear) and TPS
(non-linear) transform to visualize deformation

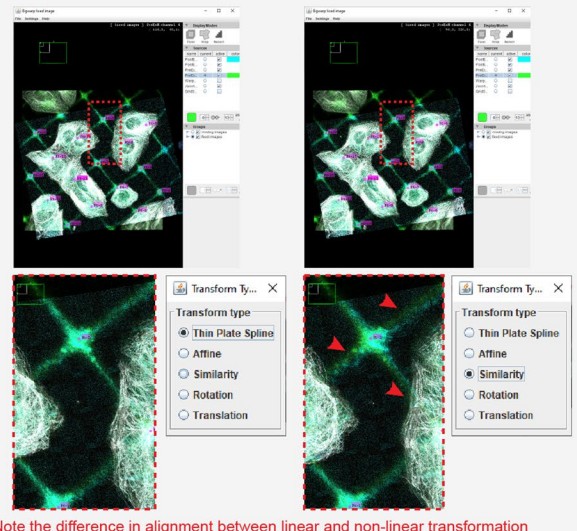

Note the difference in alignment between linear and non-linear transformation

**8.** Export corrected (non-linear TPS transformation)
moving image

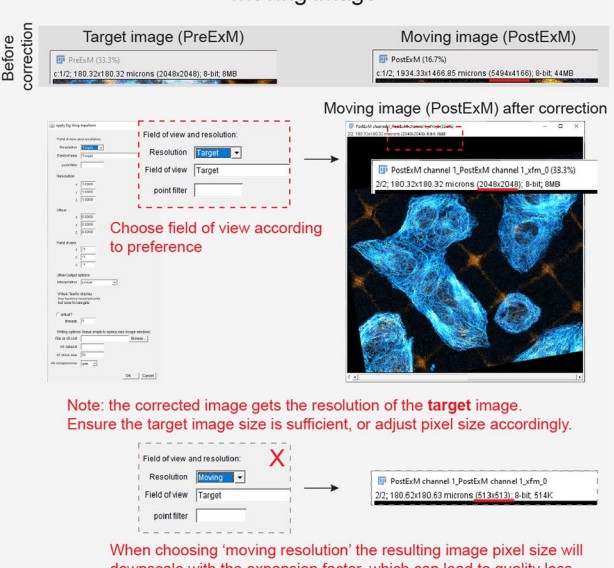

Choose field of view according
to preference

Note: the corrected image gets the resolution of the **target** image.
Ensure the target image size is sufficient, or adjust pixel size accordingly.

When choosing 'moving resolution' the resulting image pixel size will
downscale with the expansion factor, which can lead to quality loss.

**Extended Data Fig. 3 | See next page for caption.**

**Extended Data Fig. 3 | Graphical workflow for distortion correction using BigWarp. 1**) Correction can be performed in BigWarp using either a pre-expansion image or a virtual reference grid generated from the pattern design file. **2**) Correction should be performed by selecting the post-expansion image as the 'moving' image, correcting to the 'target' reference image. **3**) Correction is easier when the moving and target views are aligned, and suitable lookup tables are selected to better visualize differences between images. **4**) At least 4 corresponding landmarks (GelMap grid intersections) between the moving and target images should be selected to allow for initial alignment. **5**) Initial alignment and basic correction are performed based on the initial landmarks. The moving image will be rescaled based on the dimensions of the two images. **6**) Further correction can be performed by adjusting the viewer to allow more landmarks to be selected. **7**) Toggling between transformation techniques allows the deformation to be easily visualized in an overlay of the two images. **8**) Export the deformation-corrected non-linear thin plate spline transformation. The moving image should be exported with the resolution of the target image to avoid downscaling of the moving image due to the expansion factor.

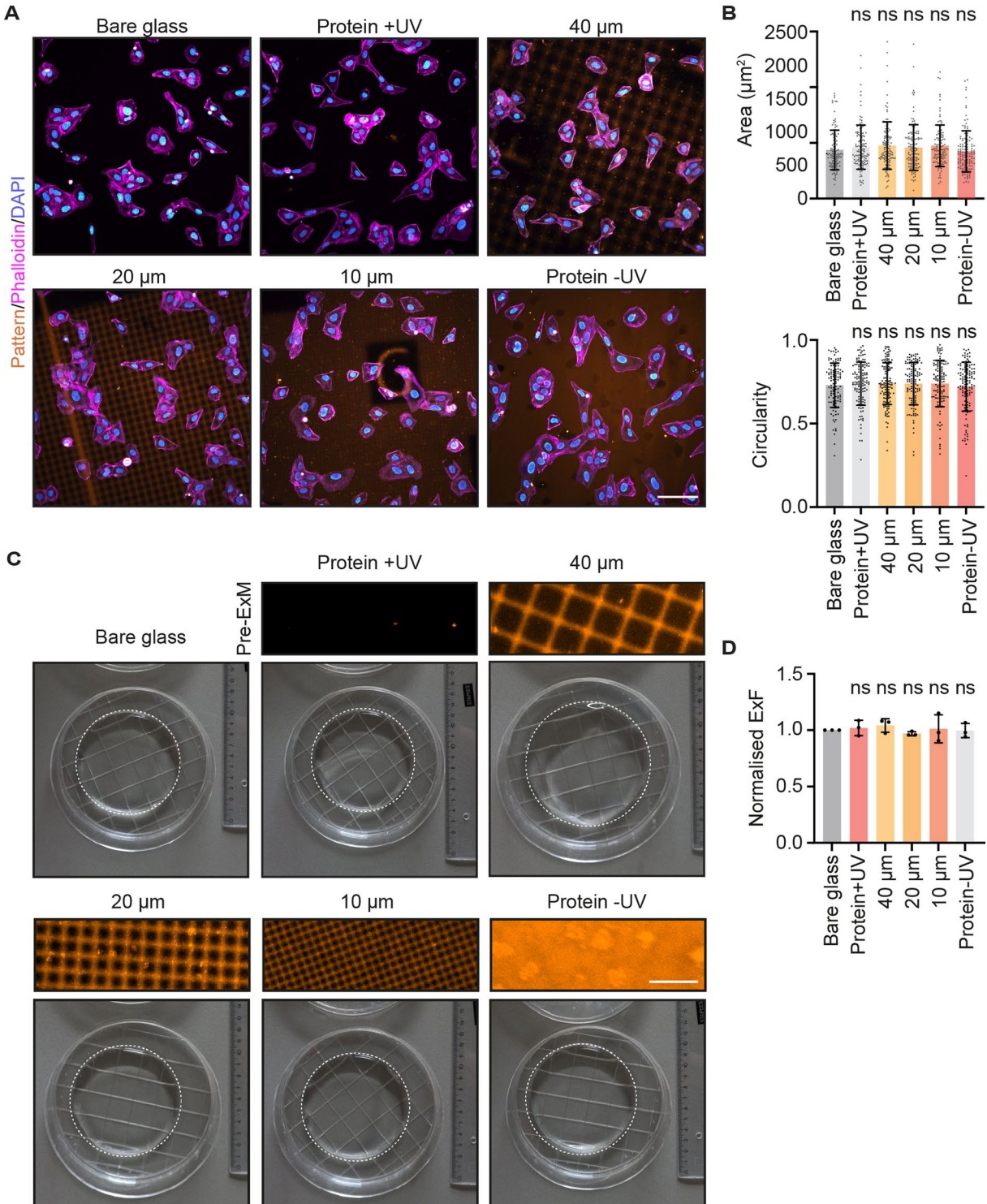

**Extended Data Fig. 4 | GelMap does not affect cellular morphology or macroscopic expansion. a**) U2OS cells grown on coverslips with increasing amount of (patterned) NBD: bare glass, coated coverslip completely illuminated with deep-UV, 40 μm grid size, 20 μm grid size, 10 μm grid size and completely coated. Representative examples from multiple fields of view in 2 biological replicates. **b**) Quantification of area and circularity of individual manually segmented cells for indicated conditions. N = 2 biological replicates, n = 110, 123, 127, 119, 114, 124 cells for: bare glass, coated coverslip completely illuminated with deep-UV, 40 μm grid size, 20 μm grid size, 10 μm grid size and completely coated, respectively. Data is represented as individual values with error bars indicating mean ± SD. **c**) Expanded gels produced using the indicated coated coverslips. **d**) Quantification of macroscopic expansion factor for indicated conditions. N = 3 replicates, error bars indicate mean ± SD. ns = not significant (Mann–Whitney U, two-tailed test). Scale bars: A: 100 μm, C: 50 μm.

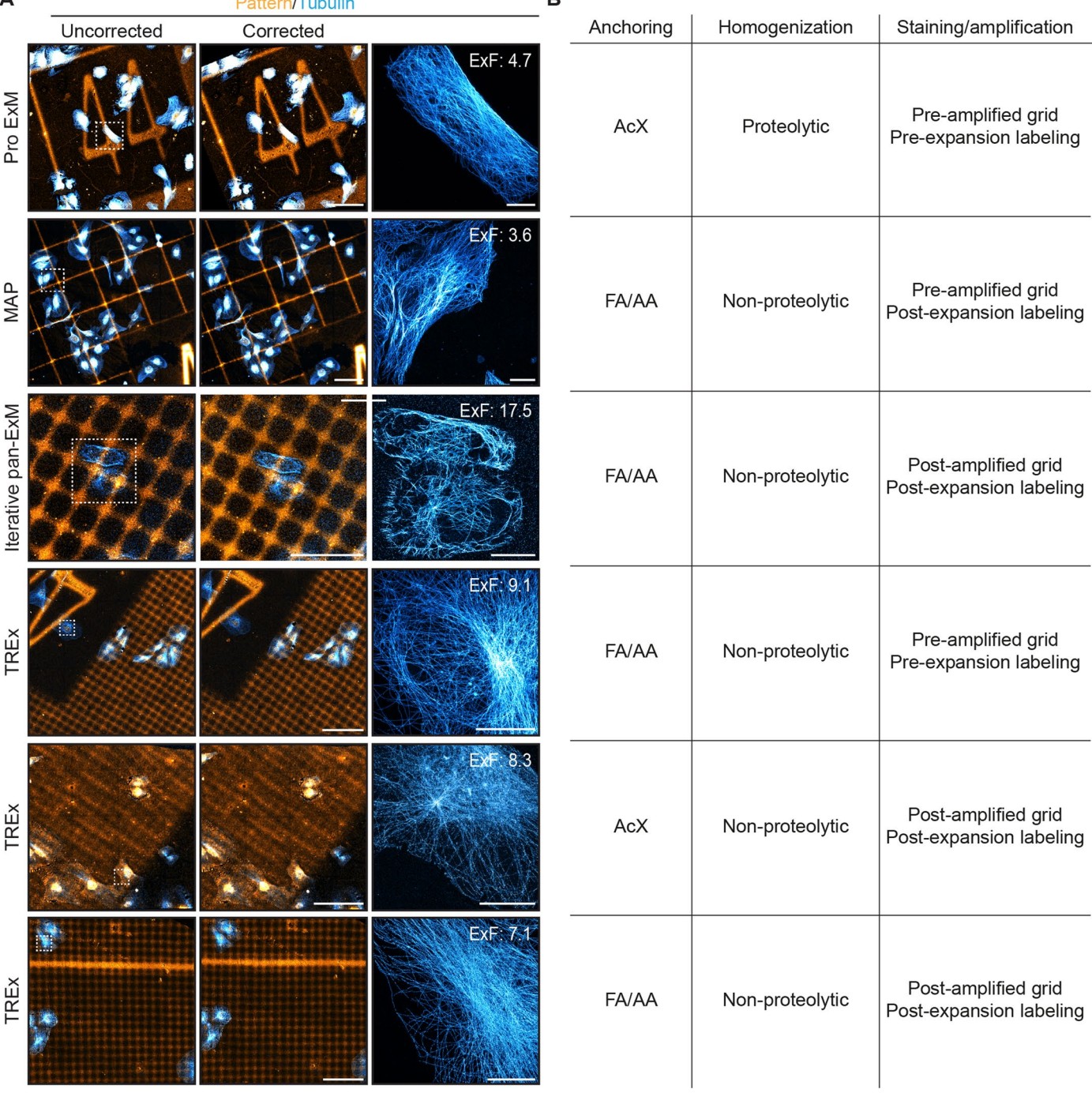

**A**

Pattern/Tubulin

**B**

| Anchoring | Homogenization | Staining/amplification |
|---|---|---|
| AcX | Proteolytic | Pre-amplified grid<br>Pre-expansion labeling |
| FA/AA | Non-proteolytic | Pre-amplified grid<br>Post-expansion labeling |
| FA/AA | Non-proteolytic | Post-amplified grid<br>Post-expansion labeling |
| FA/AA | Non-proteolytic | Pre-amplified grid<br>Pre-expansion labeling |
| AcX | Non-proteolytic | Post-amplified grid<br>Post-expansion labeling |
| FA/AA | Non-proteolytic | Post-amplified grid<br>Post-expansion labeling |

**Extended Data Fig. 5 | Compatibility of GelMap with common ExM variants. a**) Representative images of expanded cells cultured on GelMap grids and stained for tubulin (cyan) and myc (orange) using the indicated Expansion Microscopy variant and conditions. Uncorrected images show the result of linear transformation of the expanded image to a virtual reference grid, while corrected images show the result of non-linear transformation.

Representative zooms are shown together with the expansion factor measured using GelMap. **b**) Table summarizing the different anchoring, homogenization and staining/amplification strategies. Scale bars: all scale bars 100 μm, iterative pan-ExM 50 μm, all zooms 10 μm. Scale bars in expanded samples reflect pre-expansion sizes.

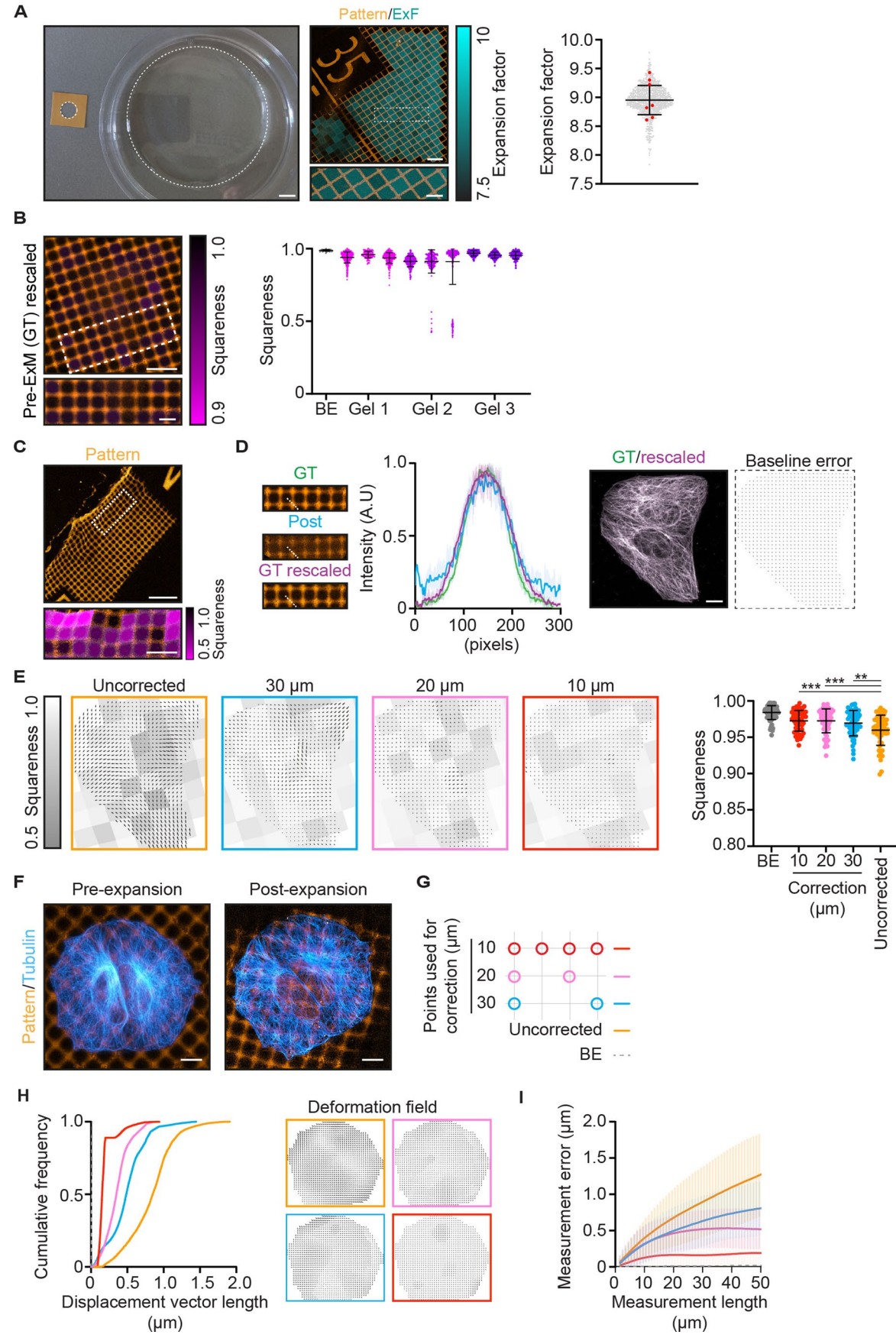

**Extended Data Fig. 6 | See next page for caption.**

**Extended Data Fig. 6 | ExM anisotropy and GelMap validation. a**) Left: example of typical macroscopic expansion factor determination with silicon mold on left and expanded gel on the right. Middle: expanded region from the same gel showing GelMap grids (orange), with the local expansion factor for each square color-coded and overlayed. Right: quantification of expansion factor of individual squares (2 regions, n = 849 squares, expansion factor 8.9 ± 0.2 (mean ± SD), gray), with macroscopic measurements from unbiased participants overlayed (n = 7 participants, expansion factor 8.9 ± 0.3 (mean ± SD), red). **b**) Left: Baseline error (BE) of the squareness quantification as determined by analysis of a resampled unexpanded grid, shown color-coded in magenta (note the same scaling of the squareness color-code as in Fig. 2a). Right: Quantification of squareness of individual squares for BE (n = 91 squares, mean ± SD squareness value of 0.99 ± 0.006) and expanded gels. 3 expanded gels, 3 regions per gel, n = 533, 116, 391, 383, 269, 756, 453, 396, 277 squares, respectively, with mean ± SD squareness values of 0.94 ± 0.04, 0.96 ± 0.02, 0.94 ± 0.03, 0.91 ± 0.04, 0.91 ± 0.08, 0.91 ± 0.2, 0.97 ± 0.02, 0.95 ± 0.02, 0.95 ± 0.03, respectively. **c**) Locally deformed region near a tear of expanded patterned grid (20 μm squares) with deformation in zoomed region color-coded in magenta as defined by squareness (expansion factor: 8.1). **d**) Baseline error determination for landmark-based registration using BigWarp. First, a non-expanded grid (GT) was taken and the average number of pixels in a square crossing was determined and compared

to that of the expanded image (Post). Next, a copy of the pre-expanded grid was resampled (GT rescaled) to the same pixel size as the post-expansion image and used for landmark-based registration to the GT image. **e**) Deformation fields from Fig. 2f overlayed with squareness analysis of the same images (left) and quantification of squareness after GelMap correction (right). n = 91, 58, 58, 58, 60 squares, respectively, with mean ± SD squareness values of 0.98 ± 0.001, 0.97 ± 0.002, 0.97 ± 0.002, 0.97 ± 0.002, 0.96 ± 0.003, respectively. **p < 0.01, ***p < 0.001 (Mann–Whitney U-test, two-tailed test), exact p values from left to right: p = 0.0005, p = 0.0001, p = 0.0074. **f**) Pre- and post-expansion images of U2OS cells cultured on GelMap coverslips (grid size: 10 μm, expansion factor: 9.2) stained for tubulin (cyan) and myc-tag (orange). **g**) Schematic of landmark spacing used for correction in E, H and I. **h**) Quantification of displacement vector length for equidistantly spaced points (1 μm) in the cellular area for uncorrected and pattern-corrected images registered to ground truth tubulin channel. Corresponding deformation fields are color-coded and shown on the right. **i**) Quantification of measurement error (deviation from expected distance assuming isotropic expansion) between pairs of points for a measurement length (mean ± SD). Scale bars: A: left, 12 mm, right, 100 μm, zoom: 40 μm. B and C: 100 μm, zoom: 40 μm, D and F: 10 μm. Scale bars in expanded samples reflect pre-expansion sizes.

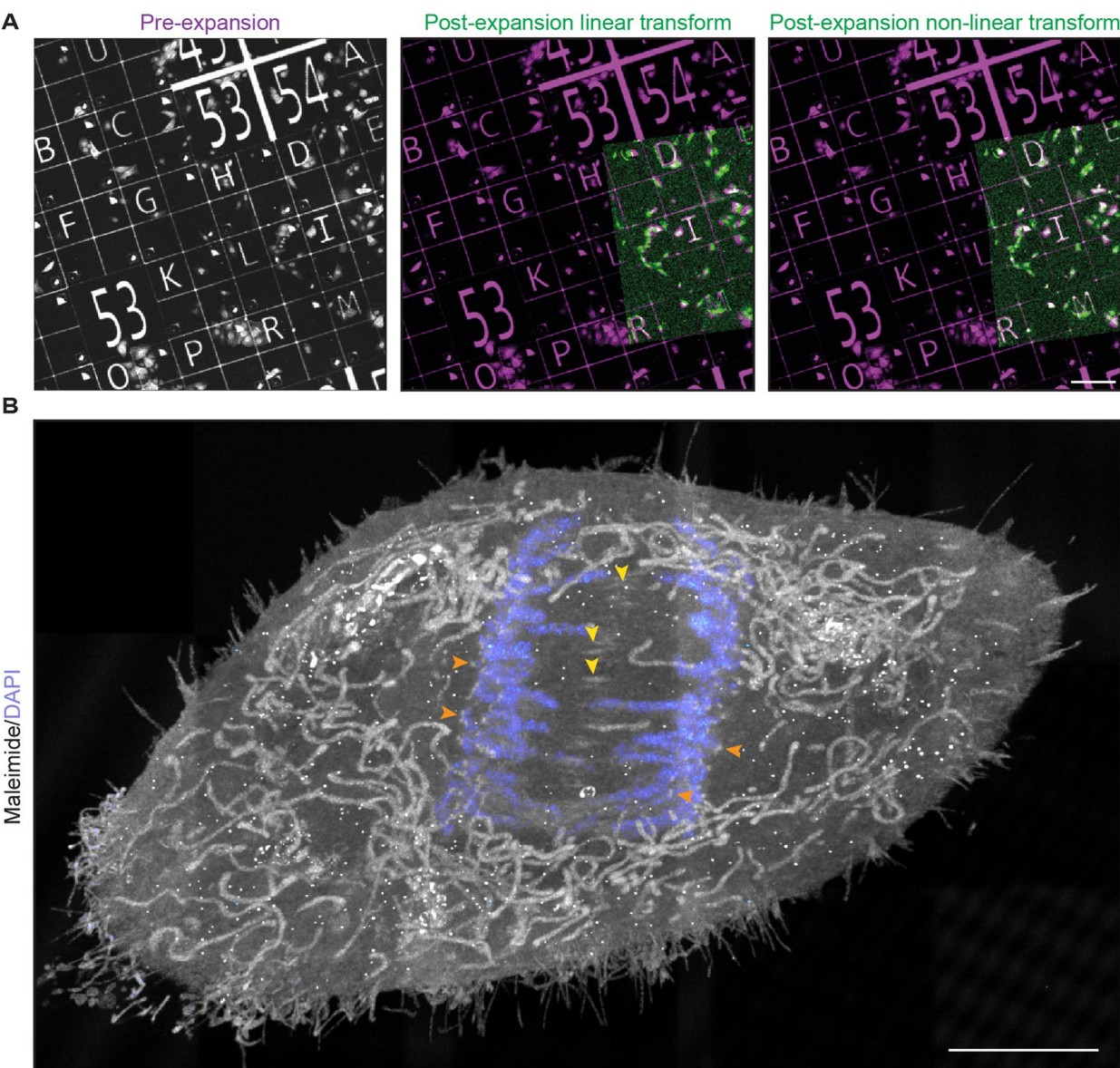

**Extended Data Fig. 7 | Correction of post-expansion images for correlative live and expansion microscopy. a)** Left: pre-expansion image of laminin-594 coated GelMap grid. Middle and right: overlayed linear and non-linear transformed post-expansion images (expansion factor: 10.7). **b)** Additional projection of mitotic cell in Fig. 3d. Scale bars: A: 200 μm B: 5 μm. Scale bars in expanded samples reflect pre-expansion sizes.

# Reporting Summary

## Statistics

For all statistical analyses, confirm that the following items are present in the figure legend, table legend, main text, or Methods section.

| n/a | Confirmed | |
|---|---|---|
| ☐ | ☒ | The exact sample size (*n*) for each experimental group/condition, given as a discrete number and unit of measurement |
| ☐ | ☒ | A statement on whether measurements were taken from distinct samples or whether the same sample was measured repeatedly |
| ☐ | ☒ | The statistical test(s) used AND whether they are one- or two-sided *Only common tests should be described solely by name; describe more complex techniques in the Methods section.* |
| ☒ | ☐ | A description of all covariates tested |
| ☒ | ☐ | A description of any assumptions or corrections, such as tests of normality and adjustment for multiple comparisons |
| ☐ | ☒ | A full description of the statistical parameters including central tendency (e.g. means) or other basic estimates (e.g. regression coefficient) AND variation (e.g. standard deviation) or associated estimates of uncertainty (e.g. confidence intervals) |
| ☐ | ☒ | For null hypothesis testing, the test statistic (e.g. *F*, *t*, *r*) with confidence intervals, effect sizes, degrees of freedom and *P* value noted *Give P values as exact values whenever suitable.* |
| ☒ | ☐ | For Bayesian analysis, information on the choice of priors and Markov chain Monte Carlo settings |
| ☒ | ☐ | For hierarchical and complex designs, identification of the appropriate level for tests and full reporting of outcomes |
| ☒ | ☐ | Estimates of effect sizes (e.g. Cohen's *d*, Pearson's *r*), indicating how they were calculated |

*Our web collection on statistics for biologists contains articles on many of the points above.*

## Software and code

Policy information about availability of computer code

| Data collection | No software was used for data collection |
|---|---|
| Data analysis | All imaging processing was done using FIJI v1.54e (Schindelin et al., 2012). Deformation correction was performed using publicly available code. The code/workflow for measuring absolute deformation and correcting is available in (Jurriens et al. 2020; doi: 10.1016/bs.mcb.2020.04.018). The code for quantifying measurement error between pairs of points for a measurement length is described in (Damstra et al. 2022; doi: 10.7554/eLife.73775), originally from (Chen et al. 2015; doi: 10.1126/science.1260088). All data visualization and statistics were performed using GraphPad Prism 9. |

For manuscripts utilizing custom algorithms or software that are central to the research but not yet described in published literature, software must be made available to editors and reviewers. We strongly encourage code deposition in a community repository (e.g. GitHub). See the Nature Portfolio guidelines for submitting code & software for further information.

## Data

Policy information about [availability of data](availability of data)

All manuscripts must include a [data availability statement](data availability statement). This statement should provide the following information, where applicable:
- Accession codes, unique identifiers, or web links for publicly available datasets
- A description of any restrictions on data availability
- For clinical datasets or third party data, please ensure that the statement adheres to our [policy](policy)

> The datasets generated and/or analyzed during the current study are available on figshare (https://doi.org/10.6084/m9.figshare.21923418). The design files used to generate photomasks for the current study are available upon request, as they require an MTA.

## Human research participants

Policy information about [studies involving human research participants and Sex and Gender in Research.](studies involving human research participants and Sex and Gender in Research.)

| | |
|---|---|
| Reporting on sex and gender | Our study did not involve human research participants |
| Population characteristics | Our study did not involve human research participants |
| Recruitment | Our study did not involve human research participants |
| Ethics oversight | Our study did not involve human research participants |

Note that full information on the approval of the study protocol must also be provided in the manuscript.

# Field-specific reporting

Please select the one below that is the best fit for your research. If you are not sure, read the appropriate sections before making your selection.

☒ Life sciences      ☐ Behavioural & social sciences      ☐ Ecological, evolutionary & environmental sciences

For a reference copy of the document with all sections, see [nature.com/documents/nr-reporting-summary-flat.pdf](nature.com/documents/nr-reporting-summary-flat.pdf)

# Life sciences study design

All studies must disclose on these points even when the disclosure is negative.

| | |
|---|---|
| Sample size | To measure expansion gel anisotropy, three regions from three separate gels were analysed. For each replicate (expansion gel FOV), the entire FOV was analysed and all data points included in analysis of expansion factor and squareness. To assess the impact of GelMap on cell morphology, two independent experiments were performed and more than 50 cells per replicate analysed. To assess the impact of GelMap on expansion factor, three independent experiments were performed. For quantification of Bassoon/Homer1 separation, 3 biological replicates were used and a total of more than 50 synapses analysed. The number of replicates for each experiment was determined by ensuring that a large sample size was obtained. |
| Data exclusions | No data points were excluded from analysis. |
| Replication | Reproducibility of correction of deformation using GelMap is demonstrated by a full replication of Fig. 2C-G, in Fig. S6F-I. |
| Randomization | There are no distinct experimental groups in this study. |
| Blinding | For Fig. S6A (measurement of macroscopic expansion factor by unbiased participants), participants were not given any indication in advance of the 'true' expansion factor. Blinding was not required for other experiments as there was no group allocation. |

# Reporting for specific materials, systems and methods

We require information from authors about some types of materials, experimental systems and methods used in many studies. Here, indicate whether each material, system or method listed is relevant to your study. If you are not sure if a list item applies to your research, read the appropriate section before selecting a response.

## Materials & experimental systems

| n/a | Involved in the study |
|---|---|
| ☐ | ☒ Antibodies |
| ☐ | ☒ Eukaryotic cell lines |
| ☒ | ☐ Palaeontology and archaeology |
| ☐ | ☒ Animals and other organisms |
| ☒ | ☐ Clinical data |
| ☒ | ☐ Dual use research of concern |

## Methods

| n/a | Involved in the study |
|---|---|
| ☒ | ☐ ChIP-seq |
| ☒ | ☐ Flow cytometry |
| ☒ | ☐ MRI-based neuroimaging |

# Antibodies

| | |
|---|---|
| Antibodies used | The following primary antibodies were used in this work: rabbit anti-laminin (1:100, Abcam ab11575), rabbit anti-myc-tag (1:100, Cell Signalling Technology 2272), rabbit anti-fibrinogen (1:100, Abcam ab34269), rat anti-tubulin YL1/2 (1:200, Abcam ab6160), mouse anti-tau (1:500, Sigma-Aldrich MAB3420), chicken anti-MAP2 (1:500, Abcam ab5392), mouse anti-Bassoon (1:250, Enzo SAP7F407), rabbit anti-Homer1 (1:250, Synaptic Systems 160003). The following secondary antibodies were used in this work: goat anti-rabbit IgG Alexa Fluor 594 (Invitrogen A-11037), goat anti-rabbit IgG Alexa Fluor 488 (Invitrogen A-11029), goat anti-rat IgG Alexa Fluor 488 (Invitrogen A-11006). goat anti-chicken IgY DyLight 488 (Invitrogen SA5-10070), goat anti-mouse IgG Alexa Fluor 647 (Invitrogen A-21236), all at a 1:250 dilution (pre-expansion labelling), or 1:500 (post-expansion labelling). |
| Validation | All primary antibodies used in this study are publicly available. We validated antibodies by confirming expected cellular localisation, in addition to the validation by the manufacturers:<br>Abcam validation (for ab11575, ab34269, ab6160, ab5392): "IHC and ICC determine whether an antibody recognizes the correct protein based on cellular and subcellular localization. Antibody specificity is confirmed by looking at cells that either do or do not express the target protein within the same tissue. Initially, our scientists will review the available literature to determine the best cell lines and tissues to use for validation. We then check the protein expression by IHC/ICC to see if it has the expected cellular localization (Figure 3). If the localization of the signal is as expected, this antibody will pass and is considered suitable for use in IHC/ICC.".<br>Cell Signalling Technology validation (for 2272): "All CST™ antibodies that are approved for use in immunofluorescent assays have undergone a rigorous validation process.Validation Steps Include: Cell lines or tissues with known target expression levels are used to verify specificity. Appropriate cell lines and tissues are used to verify subcellular localization. Antibody performance is assessed on appropriate tissues. Cells are subjected to phosphatase treatment to verify phospho-specificity. Target specificity is also verified with the use of known knockout or null cell lines. Cells are subjected to siRNA treatment or over-expression of the target protein to verify target specificity. Activation state specification, target expression, and translocation are examined using ligands or inhibitors to modulate pathway activity. Requirement of threshold signal-to-noise ratio in antibody:isotype comparison and minimum fold-induction for phospho-specific antibodies ensures the greatest possible sensitivity. Fixation and permeabilization conditions are optimized; alternative protocols are recommended if necessary. Stringent testing ensures lot-to-lot consistency."<br>Sigma-Aldrich validation (for MAB3420): "Immunohistochemistry: Routinely tested on rat brain tissue."<br>Enzo validation (for SAP7F407): "Purified from hybridoma tissue culture supernatant. Protein G affinity purified."<br>Synaptic Systems validation (for 160003): "We at Synaptic Systems aim to characterize and validate our products as best as possible to provide antibodies of highest quality and reliability. All our antibodies are produced in-house and therefore we have full control over batch testing and quality control. Specific for Homer 1. Cross-reactivity of the serum to Homer 2 and 3 was removed by pre-adsorption with Homer 2 (aa 1 - 176) and Homer 3 (aa 1 - 177))." |

# Eukaryotic cell lines

Policy information about cell lines and Sex and Gender in Research

| | |
|---|---|
| Cell line source(s) | WT U2OS cells were obtained from ATCC. U2OS cells expressing YFP-H2B and mCherry-tubulin were a gift from from Jonne Raaijmakers and René Medema, NKI Amsterdam |
| Authentication | Cells were authenticated by ATCC, by confirming the STR profile of U2OS. |
| Mycoplasma contamination | Cells were routinely checked for mycoplasma and discarded if positive. |
| Commonly misidentified lines (See ICLAC register) | No commonly misidentified cell lines were used in the study. |

# Animals and other research organisms

Policy information about studies involving animals; ARRIVE guidelines recommended for reporting animal research, and Sex and Gender in Research

| | |
|---|---|
| Laboratory animals | 10-month-old, TRAP2 heterozygous mice (Jax #030323) were used. Up until the moment of perfusion, mice were group-housed (2–4 per cage) in a temperature- and humidity-controlled room (22 ± 2°C and 60–65% respectively) under a 12 h light/dark cycle (lights on at 7am) with ad libitum access to water and standard laboratory chow [Special Diet Services [SDS], product code CRM(E)]. |
| Wild animals | No wild animals were used in the study. |

| | |
|---|---|
| Reporting on sex | Sex was not considered in sample collection; male mice were used for tissue slices in Fig.4 |
| Field-collected samples | No field collected samples were used in the study. |
| Ethics oversight | All animal experiments were carried out according to the regulations of Utrecht University and in agreement with Dutch law (Wet op de Dierproeven, 1996) and European regulations (Directive 2010/63/EU). |

Note that full information on the approval of the study protocol must also be provided in the manuscript.

