## [Peer Review File · Nature Methods]

Peer Review Information

Manuscript Title: GelMap: Intrinsic calibration and deformation mapping for expansion microscopy

Corresponding author name(s): Lukas Kapitein

Editorial Notes: n/a

Reviewer Comments & Decisions:

Decision Letter, initial version:

Dear Lukas,

Your Article, "GelMap: Intrinsic calibration and deformation mapping for expansion microscopy", has now been seen by three reviewers. As you will see from their comments below, although the reviewers find your work of considerable potential interest, they have raised a number of concerns. We are interested in the possibility of publishing your paper in Nature Methods, but would like to consider your response to these concerns before we reach a final decision on publication.

We therefore invite you to revise your manuscript to address these concerns. We think in the revised version you should emphasize the main approach using a gel-embedded reference grid rather than the approach involving photoactivation, as I think the latter generated more skepticism than interest from the reviewers. You can keep this in the paper, but please remove any claims about practical performance and performance for correcting distortions in 3D. If you want to make any claims about performance in 3D, a demonstration on a biological sample expanded to at least 4x will be necessary. We will not require this for your paper.

As far as technical concerns, the referees were not convinced that the approach is generally applicable, and these concerns sort of fell along three lines, the first being applicability to different expansion protocols, the second, related point being applicability to very highly expanded samples, and the third being application to more challenging tissues.

For the first two points, we think these should be addressed experimentally by showing the approach is applicable to a few types of commonly used ExM protocols, including at least one iterative protocol.

Even if the results are not particularly good for iterative expansion, this will be of interest to potential users.

With regards to more challenging tissues (ref 3), we see the value this could add but will not require it. It should at least be discussed.

We think the other technical concerns and clarifications are reasonable and straightforward, but we are committed to providing a fair and constructive peer-review process. Do not hesitate to contact us if there are specific requests from the reviewers that you believe are technically impossible or unlikely to yield a meaningful outcome. If you have any questions about our expectations for the revision, please feel free to email me.

[Redacted] This URL links to your confidential home page and associated information about manuscripts you may have submitted, or that you are reviewing for us. If you wish to forward this email to co-authors, please delete the link to your homepage.

We hope to receive your revised paper within three months. If you cannot send it within this time, please let us know. In this event, we will still be happy to reconsider your paper at a later date so long as nothing similar has been accepted for publication at Nature Methods or published elsewhere.

OPEN SCIENCE REQUIREMENTS

REPORTING SUMMARY AND EDITORIAL POLICY CHECKLISTS

IMAGE INTEGRITY

DATA AVAILABILITY

All novel DNA and RNA sequencing data, protein sequences, genetic polymorphisms, linked genotype and phenotype data, gene expression data, macromolecular structures, and proteomics data must be deposited in a publicly accessible database, and accession codes and associated hyperlinks must be provided in the “Data Availability” section.

Please include a “Data availability” subsection in the Online Methods. This section should inform readers about the availability of the data used to support the conclusions of your study, including accession codes to public repositories, references to source data that may be published alongside the paper, unique identifiers such as URLs to data repository entries, or data set DOIs, and any other statement about data availability. At a minimum, you should include the following statement: “The data that support the findings of this study are available from the corresponding author upon request”, describing which data is available upon request and mentioning any restrictions on availability. If DOIs are provided, please include these in the Reference list (authors, title, publisher (repository name), identifier, year). For more guidance on how to write this section please see: <http://www.nature.com/authors/policies/data/data-availability-statements-data-citations.pdf>

CODE AVAILABILITY

Please include a “Code Availability” subsection in the Online Methods which details how your custom code is made available. Only in rare cases (where code is not central to the main conclusions of the paper) is the statement “available upon request” allowed (and reasons should be specified).

MATERIALS AVAILABILITY

SUPPLEMENTARY PROTOCOL

To help facilitate reproducibility and uptake of your method, we ask you to prepare a step-by-step Supplementary Protocol for the method described in this paper. We [encourage authors to share their step-by-step experimental protocols](https://www.nature.com/nature-research/editorial-policies/reporting-standards#protocols) on a protocol sharing platform of their choice and report the protocol DOI in the reference list. Nature Portfolio's Protocol Exchange is a free-to-use and open resource for protocols; protocols deposited in Protocol Exchange are citable and can be linked from the published article. More details can found at www.nature.com/protocolexchange/about.

ORCID

Nature Methods is committed to improving transparency in authorship. As part of our efforts in this direction, we are now requesting that all authors identified as 'corresponding author' on published papers create and link their Open Researcher and Contributor Identifier (ORCID) with their account on the Manuscript Tracking System (MTS), prior to acceptance. This applies to primary research papers

only. ORCID helps the scientific community achieve unambiguous attribution of all scholarly contributions. You can create and link your ORCID from the home page of the MTS by clicking on 'Modify my Springer Nature account'. For more information please visit www.springernature.com/orcid.

Sincerely,
Rita

Rita Strack, Ph.D.
Senior Editor
Nature Methods

Reviewers' Comments:

Reviewer #1:

Remarks to the Author:

In the study, Darmstra, Passmore and colleagues described a new workflow to calibrate gel deformation for expansion microscopy (ExM). The method, termed GelMap, is achieved by photo-printing and anchoring a series of fluorescence fiducial-grid-patterns onto the bottom of the fixed samples. After the ExM treatment, the pre/post expansion images of the grids are compared to determine the deformation and expansion factors. GelMap simplified the image registration in ExM and can be used as a benchmark in ExM protocol development.

The design of the work is mind-blowing and also well explained, but there are still some general questions that need to be clarified:

1. The authors examined two different fluorophore-anchoring strategies: 1) direct photo-printing of fluorescent protein onto the coverslip, then culture and fix cells/ tissues onto the coverslip. During the ExM procedure, the grid-proteins/antibodies were crosslinked and expanded along with the sample. 2) immerse the sample with a photo-activatable rhodamine-AcX and co-polymerize the dye into the hydrogel. After the gelation, activate the fluorophore using UV laser at specified locations, then expand

the gel. Here the first approach was used through the text, while the second approach was only mentioned in Fig S1F, with a proof of concept experiment. But at 1.5 x expansion, the resulting signals were already insufficient, so it's not convincing to use this approach in 4-20x expanded gels or in 3D. This is a major limitation and would require more input.

Except for increasing photo-uncaged fluorophore density in the monomer solution, it might be reasonable to find another photo-activatable molecule that can also be post-magnified in fluorescence. One possible approach might be the photo-cleavable DNA with hybridization-chain-reaction. (Lin R et al. Nature Methods. 2018. doi:10.1038/nmeth.4611)

2. The description of the above mentioned photo-printing/activating methods are not sufficient to be reproduced elsewhere. The microprinting technique (mask-assist photopatterning) mentioned in the reference (Théry, 2010, the only reference on photo-printing) requires an anti-fouling polymer (such as PEG) coated coverslip, with deep UV illumination, the PEG chain is partially cleaved and activated to achieve the following protein (ECM) adsorption. Alternatively, directly photobleaching on fluorophore can also be create free radicals so that the protein in solution would react and bind to the coverslip on designated locations. Oxygen in the solution will therefore influence the binding affinity/ final protein density on the surface. However both differ to the method described in this work (line 406-417): UV illumination was applied directly to a dried ECM coated coverslip. This mechanism is not clear to me. References or some descriptions could help to clarify.

It is also necessary to provide a detailed protocol, including complete procedures, the settings of UV illumination (line 413, brand, model, power density etc), the settings for UV activation (line 432, laser power, time duration, how to achieve the patterns on the microscope or with UV lamp) and one or more research article references in the maintext referring to the methodology of the photo-printing and the photo-activation molecule. Since the precision of the printed patterns in the pre-ExM images were not shown quantitatively, the reference should also contain information on the accuracy and reproducibility of such micro-patterning techniques.

3. Following the photo-printing question, although cells attach to the surface 'irrespective of the underlying protein grid', the modification on the surface may still alter cell morphology. Could the authors perform a simple experiments with stress fiber stained cells on the GelMap coverslips. Grids range from 10 μm x 10 μm to 100 μm x 100 μm . Normal fluorescence microscopy on the fixed samples would be sufficient.

4. The squareness was well shown as a local distortion marker, and the distortion correction based on the grids was also successfully established here. However, I feel the link between the two factors is missing, possibly due to that the TReX is developed to have low distortion. For a broadly application and examination, some of the other well-known and widely used protocol such as ExM, MAP and U-ExM should be tested. The MAP was already shown to have different expansion factor globally as a gel or at nanoscale for some organelles. It might be interesting to see by applying the correction with GelMap, if

the local distortion can also be corrected, or when the squareness is under a certain value, the registration can not reveal the ground truth or correlate the squareness to specific sub-cellular structures that caused local distortion.

Reviewer #2:

Remarks to the Author:

Expansion microscopy is a super-resolution method that consists in embedding biological samples, proteins, cells, or tissues, in a swellable polymer that can expand with different expansion factors depending on the method (4x, 9x, or more). Despite many demonstrations that the expansion is isotropic, there are still local deformations that can occur during the expansion. Also, the expansion factor can vary between gels and even within the same gel. To overcome this problem, Damstra, Passamore, et al, have developed an approach to incorporate a grid in the expansion gel in order to correct local deformations and measure precisely the expansion factor. Furthermore, the authors demonstrate that this technique can be used for correlative live imaging/expansion microscopy. This development is very useful for the rapidly growing community that lacks such a tool to better measure expansion factors. The experiments are very well done and very convincing, nevertheless I think that some points must be clarified :

- It seems to me that only the TReX method has been used here but many variants of the original ExM protocol exist. How compatible is this approach with pro-ExM or MAP? For example here the protocol uses enzymatic digestion, but is it compatible with the thermal denaturation that replaces enzymatic digestion in the MAP protocol? Similarly, the authors use acryloyl, do other anchoring molecules work with this approach?

- Is the grid preserved in iterative approaches (for Pan-ExM for example)?

- For the live/ExM correlative imaging (which is very well executed), the authors chose the centriole as a molecular ruler to verify that their expansion factor is correct. However, they used a maleimide label, a dye poorly characterized for the centriole (which looks small for 9X expansion). Can the authors give the same quantification with tubulin or acetylated tubulin?

- The registration between the grids before and after expansion is done with BiGWarp under imageJ. This is a good solution that will easily allow other labs to use it. Nevertheless, I found that the material and method were not very clear. Would it be possible to have a graphical version of the steps in supplemental data? And does BigWARP work in dual color, or triple color?

- It seems that the signal of the grid after expansion is really weak, is that why the authors used a moderate expansion of 3X? Is it necessary to use systematically the staining with antibody for 9X? Also for photoactivation, the signal is very weak at 1.5X, what about at 9X? Would it be possible to have an idea of the intensity of the signal according to the expansion factor?

Minor points :

- Can the authors give a more precise reference for the Roche laminin?

- The photomask designs are very well done, will they be available in the final version (I did not find them here)?

- In the intro, the authors quote that the ExM field uses nuclear pores as a molecular ruler. I haven't seen this used yet, can they cite the article in question?

Reviewer #3:

Remarks to the Author:

The authors present a new method, termed GelMap, introducing a fluorescent grid that can integrate with an expandable hydrogel and expand with biological samples. With the grid, they impressively demonstrate reference-free correction of deformations accompanied by expansion. GelMap also provides assistance with sample navigation and facilitates to precisely determine expansion factors for calibration.

The paper is clearly written and presents well-organized data. In my opinion, it is an original approach that offers a valuable tool for setting up an expansion protocol or developing a new one. However, this work needs to clarify further the scope of application and degree of profit.

1) Although this work includes impeccable demonstrations with cultured cells, the manuscript shows minimal use of GelMap with tissue samples. Tissue samples, particularly when they are thick (typically 0.5 mm after expansion or thicker), are important targets when applying expansion microscopy, since super-resolution microscopic techniques are often unfeasible. The authors should include results with common tissue sample types, such as tissue specimens expressing fluorescent proteins and/or those immunolabeled with fluorescent antibodies.

More critically, a supplemental protocol introduced for the calibration and deformation correction of expanded tissue images would be beneficial when it enhances the precise measurement of structures in 3D. The current work includes a conceptual result using a photoactivatable dye integrated in a hydrogel with no biological sample, but the experiment in Figure S1F seems to have been carried out in 2D. In principle, two-photon lithography can form a 3D grid in a tissue-hydrogel hybrid. However, considering the necessary equipment and technical difficulties, particularly regarding how to secure a sufficient

signal level by 3D lithography because the signal drop in Figure S1F is already remarkable after 1.5x expansion, the authors should suggest a practical 3D GelMap protocol with 3D deformation analysis and correction that users would be willing to choose instead of 3D calibration based on pre-expansion images.

2) The manuscript only includes results generated with TReX. Since the authors conclude that GelMap is anticipated to be compatible with many existing and new expansion protocols, it would be helpful to demonstrate to the reader using other well-established protocols. In particular, the merit of using GelMap will be maximal when a sample is post-expansion immunolabeled and reference-free calibration is necessary. MAP must be a representative method in this class, and a revised protocol from the same group (eMAP, a non-iterative protocol; Park J et al. Science Advances 2021) might be a good candidate to demonstrate this.

3) The authors included many rigorous and careful works on deformation analysis. I recommend a couple of additional analyses to further improve the manuscript. The effects of grid proteins on expansion factors and deformation (e.g., no effects, suppressing gross deformation, lowering the expansion factor, or causing deformation along grid lines) might be of interest, and such information will be considered when users decide to exclude the use of grids in their expansion workflow once they successfully establish the protocol due to GelMap.

The squareness and local expansion factor (Figs. 2A and S2A,B) must be determined not only by expansion-related deformations but also by an analysis process (e.g., corner detection precision in pixelated images). Please provide the baseline error levels observed before expansion.

4) The downsides and limitations of using GelMap have not been properly discussed in the manuscript. By using a fluorescent grid, the number of fluorescence channels for microscopy should decrease by one. When an antibody is used to label grid proteins, the host species should be excluded from the antibody pool for immunolabeling of target proteins. It would be helpful for the reader if the additional steps and the cost of using GelMap for the initial setup and routine usage were summarized in the manuscript. The signal level of a grid is, again, of concern. In Figures S3A, 2C, and 4B,C, grid signals are found to noticeably drop after expansion, while other channels (e.g., tubulin) seem relatively stable. It would be helpful to provide a guideline for the necessary grid signal level for successful calibration.

Minor points:

- 1) Figure S2A,C: please indicate beside the color bars how the colors match the values.
- 2) It is unclear how to interpret the data in Figure 2C,E–G and those in Figure S2D–G differently.
- 3) Please improve the Materials and Methods section, which contains incomplete/unclear sentences, undefined acronyms, and missing catalog numbers.

Point-to-point response for:

GelMap: Intrinsic calibration and deformation mapping for expansion microscopy

Hugo Damstra, Josiah Passmore et al.

Reviewer #1

In the study, Damstra, Passmore and colleagues described a new workflow to calibrate gel deformation for expansion microscopy (ExM). The method, termed GelMap, is achieved by photo-printing and anchoring a series of fluorescence fiducial-grid-patterns onto the bottom of the fixed samples. After the ExM treatment, the pre/post expansion images of the grids are compared to determine the deformation and expansion factors. GelMap simplified the image registration in ExM and can be used as a benchmark in ExM protocol development.

The design of the work is mind-blowing and also well explained, but there are still some general questions that need to be clarified:

➤ **We thank the reviewer for the kind words and constructive feedback. In the revised manuscript we have addressed all questions, as detailed below.**

1. The authors examined two different fluorophore-anchoring strategies: 1) direct photo-printing of fluorescent protein onto the coverslip, then culture and fix cells/ tissues onto the coverslip. During the ExM procedure, the grid-proteins/antibodies were crosslinked and expanded along with the sample. 2) immerse the sample with a photo-activatable rhodamine-AcX and co-polymerize the dye into the hydrogel. After the gelation, activate the fluorophore using UV laser at specified locations, then expand the gel. Here the first approach was used through the text, while the second approach was only mentioned in Fig S1F, with a proof of concept experiment. But at 1.5 x expansion, the resulting signals were already insufficient, so it's not convincing to use this approach in 4-20x expanded gels or in 3D. This is a major limitation and would require more input.

Except for increasing photo-uncaged fluorophore density in the monomer solution, it might be reasonable to find another photo-activatable molecule that can also be post-magnified in fluorescence. One possible approach might be the photo-cleavable DNA with hybridization-chain-reaction. (Lin R et al. Nature Methods. 2018. doi:10.1038/nmeth.4611)

➤ **Indeed, most of our manuscript is built around the photolithography approach, because this approach is straightforward, user-friendly and adoptable, and will suffice for most applications. We included the photo-activation method to illustrate a potential approach for mapping expansion factors and deformations in three dimensions. We agree that the specific approach we chose for photo-activation results in substantial loss of intensity. We think this is due to the limited density of fluorescent molecules attached to the gel and the lack of signal amplification. Following the editorial guidance, we have now reduced the emphasis on the photo-uncaging approach and removed any claims about practical performance, but instead present it as a potential avenue for three-dimension deformation mapping. We have now also included the excellent suggestion from the reviewer to use photo-cleavable DNA to enable amplification and might try this out in future work.**

2. The description of the above mentioned photo-printing/activating methods are not sufficient to be reproduced elsewhere. The microprinting technique (mask-assist photopatterning) mentioned in the reference (Théry, 2010, the only reference on photo-printing) requires an anti-fouling polymer (such as PEG) coated coverslip, with deep UV illumination, the PEG chain is partially cleaved and activated to achieve the following protein (ECM) adsorption. Alternatively, directly photobleaching on fluorophore can also be create free radicals so that the protein in solution would react and bind to the coverslip on designated locations. Oxygen in the solution will therefore influence the binding affinity/ final protein density on the surface. However both differ to the method described in this work (line 406-

417): UV illumination was applied directly to a dried ECM coated coverslip. This mechanism is not clear to me. References or some descriptions could help to clarify.

It is also necessary to provide a detailed protocol, including complete procedures, the settings of UV illumination (line 413, brand, model, power density etc), the settings for UV activation (line 432, laser power, time duration, how to achieve the patterns on the microscope or with UV lamp) and one or more research article references in the main text referring to the methodology of the photo-printing and the photo-activation molecule. Since the precision of the printed patterns in the pre-ExM images were not shown quantitatively, the reference should also contain information on the accuracy and reproducibility of such micro-patterning techniques.

- We cited Thery et al as an early example of using photolithography in biological research that inspired our approach, but understand this can lead to confusion because we use a simplified version of this approach. In the cited work, the goal is to create specific areas where cells can adhere. Therefore, coverslips are pre-coated with PEG to block adhesion, after which the PEG is burnt away in specific places to facilitate patterned cell adhesion. In our case we do not use PEG, because we do not seek to pattern cells or prevent their adhesion. Instead, we coat the coverslips uniformly with a protein (NBD or Laminin) and then burn it away in specific locations using the photomask to create our GelMap grids. To clarify our approach and facilitate replication in other labs, we have added Supplemental Figure 2, which graphically outlines our patterning approach. We have also augmented the method sections with more details about the used equipment and settings.

3. Following the photo-printing question, although cells attach to the surface 'irrespective of the underlying protein grid', the modification on the surface may still alter cell morphology. Could the authors perform a simple experiments with stress fiber stained cells on the GelMap coverslips. Grids range from $10\ \mu\text{m} \times 10\ \mu\text{m}$ to $100\ \mu\text{m} \times 100\ \mu\text{m}$. Normal fluorescence microscopy on the fixed samples would be sufficient.

- To quantify the effect of the grids on cell morphology, we compared cell morphology of cells grown on bare glass, on glass uniformly coated with protein, on glass with different GelMap grids (10, 20, 40 μm), and on glass uniformly coated with protein that was subsequently exposed to UV light without a photomask to completely remove the protein coating. The results are shown in Supplemental Figure 4A and reveal that cell morphology is not altered by the patterned proteins. In case researchers will find that their specific cell type is sensitive to the patterned protein, it is always possible to overlay the GelMap grids with another protein, as we demonstrate for cultured dissociated neurons by overlaying NBD GelMap grids with a mix of PLL and laminin (Fig. 3A).

4. The squareness was well shown as a local distortion marker, and the distortion correction based on the grids was also successfully established here. However, I feel the link between the two factors is missing, possibly due to that the TReX is developed to have low distortion. For a broadly application and examination, some of the other well-known and widely used protocol such as ExM, MAP and U-ExM should be tested. The MAP was already shown to have different expansion factor globally as a gel or at nanoscale for some organelles. It might be interesting to see by applying the correction with GelMap, if the local distortion can also be corrected, or when the squareness is under a certain value, the registration can not reveal the ground truth or correlate the squareness to specific sub-cellular structures that caused local distortion.

- To examine the applicability of GelMap for other variants of Expansion Microscopy, we have now tested our approach using four commonly used ExM variants: pro-ExM, MAP, TReX, and pan-ExM, an iterative expansion approach. The results are shown in Supplemental Figure 5 and reveal that GelMap can be readily used in a wide variety of different expansion variants. Furthermore, to probe the correlation between squareness and the field deformation obtained from landmark-based registration, we now included a panel that shows the overlay between squareness and the deformation field as determined using landmark-based registration, see Supplemental Figure 6E. These data demonstrate that regions with reduced squareness also show more deformations in the deformation field. Finally, we now show that correcting images using the landmark-based registration also improves the squareness metric.

Reviewer #2

Expansion microscopy is a super-resolution method that consists in embedding biological samples, proteins, cells, or tissues, in a swellable polymer that can expand with different expansion factors depending on the method (4x, 9x, or more). Despite many demonstrations that the expansion is isotropic, there are still local deformations that can occur during the expansion. Also, the expansion factor can vary between gels and even within the same gel. To overcome this problem, Damstra, Passamore, et al, have developed an approach to incorporate a grid in the expansion gel in order to correct local deformations and measure precisely the expansion factor. Furthermore, the authors demonstrate that this technique can be used for correlative live imaging/expansion microscopy. This development is very useful for the rapidly growing community that lacks such a tool to better measure expansion factors. The experiments are very well done and very convincing, nevertheless I think that some points must be clarified :

➤ **We thank the reviewer for the kind words regarding our manuscript. In the revised manuscript, we have addressed all comments of the reviewer.**

1. It seems to me that only the TReX method has been used here but many variants of the original ExM protocol exist. How compatible is this approach with pro-ExM or MAP? For example here the protocol uses enzymatic digestion, but is it compatible with the thermal denaturation that replaces enzymatic digestion in the MAP protocol? Similarly, the authors use acryloyl, do other anchoring molecules work with this approach?

➤ **To examine the applicability of GelMap for other variants of Expansion Microscopy, we have now tested our approach using four commonly used ExM variants: pro-ExM, MAP, TReX, and pan-ExM, an iterative expansion approach. As such, we have also used different denaturation approaches (i.e. enzymatic digestion and thermal denaturation) and different anchoring approaches (i.e. AcX and FA/AA). The results are shown in Supplemental Figure 5 and reveal that GelMap can be readily used in a wide variety of different expansion approaches.**

2. Is the grid preserved in iterative approaches (for Pan-ExM for example)?

➤ **As shown in the new Supplemental Figure 5, the grid is preserved when using pan-ExM, an iterative expansion approach.**

3. For the live/ExM correlative imaging (which is very well executed), the authors chose the centriole as a molecular ruler to verify that their expansion factor is correct. However, they used a maleimide label, a dye poorly characterized for the centriole (which looks small for 9X expansion). Can the authors give the same quantification with tubulin or acetylated tubulin?

➤ **In the correlative example, we could use the pre-expanded and post-expanded images of the cell to verify that the expansion factor determined by GelMap was correct (see also Supplemental Figure 7). To our satisfaction, the estimated expansion factor was consistent with the observed size of the centriole that happened to be visible in the total protein stain. As most ExM work on centrioles has been performed with different fixation and anchoring strategies than we used in our figure, we felt that it would be outside the scope of the current manuscript to optimize centriole stainings for correlative live/ExM imaging. To provide an alternative validation for a different and well-characterized structure, the revised manuscript now includes new data from tissue where we have quantified the spacing between the presynaptic protein Bassoon and the postsynaptic protein Homer, which revealed a spacing of 117 nm consistent with earlier results from other labs (see paper for references).**

4. The registration between the grids before and after expansion is done with BiGWarp under imageJ. This is a good solution that will easily allow other labs to use it. Nevertheless, I found that the material and method were not very clear. Would it be possible to have a graphical version of the steps in supplemental data? And does BigWARP work in dual color, or triple color?

➤ **We have rewritten the material and method section for increased clarity and we have added a graphical workflow for the use of Big Warp for corrections, shown in Supplemental Figure 3. Big Warp is indeed compatible with multichannel images.**

5. *It seems that the signal of the grid after expansion is really weak, is that why the authors used a moderate expansion of 3X? Is it necessary to use systematically the staining with antibody for 9X? Also for photoactivation, the signal is very weak at 1.5X, what about at 9X? Would it be possible to have an idea of the intensity of the signal according to the expansion factor?*

- In Figure 1B we used an expansion factor of 3, because that enabled us to image the entire gel in one piece. In all biological examples we use an expansion factor of 8-11, where we indeed use antibody labeling to amplify the better visualize the GelMap grids. In the revised manuscript, we demonstrate that GelMap grids are visible in gels obtained with different anchoring, homogenization and expansion recipes, even when using 17-fold iterative expansion (Supplemental Figure 5). However, we also observed that the level of signal retention depends on the exact anchoring and homogenization strategy used. For example, post-expansion amplification of the grid works better in combination with paraformaldehyde/acrylamide anchoring, compared to AcX-based anchoring. We have commented on these observations in the revised manuscript.

Following the editorial guidance, we have now reduced the emphasis on the photo-activation approach, where signal reduction is indeed a problem. However, in the text we included the excellent suggestion from reviewer 1 to use photo-cleavable DNA in order to enable amplification.

Minor points:

6. *Can the authors give a more precise reference for the Roche laminin?*

- We have included a more precise reference in the text.

7. *The photomask designs are very well done, will they be available in the final version (I did not find them here)?*

- Due to our pending patent application sharing photomasks design files require an MTA but will be available upon request, as per recommendation of the Utrecht University Technology Transfer office.

8. *In the intro, the authors quote that the ExM field uses nuclear pores as a molecular ruler. I haven't seen this used yet, can they cite the article in question?*

- We have added the reference in the text (Louvel et al, BioRxiv) and reformulated the text to not suggest this is common practice in the field.

Reviewer #3

The authors present a new method, termed GelMap, introducing a fluorescent grid that can integrate with an expandable hydrogel and expand with biological samples. With the grid, they impressively demonstrate reference-free correction of deformations accompanied by expansion. GelMap also provides assistance with sample navigation and facilitates to precisely determine expansion factors for calibration.

The paper is clearly written and presents well-organized data. In my opinion, it is an original approach that offers a valuable tool for setting up an expansion protocol or developing a new one. However, this work needs to clarify further the scope of application and degree of profit.

➤ **We thank the reviewer for the kind words. In the revised manuscript, we have addressed all comments of the reviewer.**

1, Although this work includes impeccable demonstrations with cultured cells, the manuscript shows minimal use of GelMap with tissue samples. Tissue samples, particularly when they are thick (typically 0.5 mm after expansion or thicker), are important targets when applying expansion microscopy, since super-resolution microscopic techniques are often unfeasible. The authors should include results with common tissue sample types, such as tissue specimens expressing fluorescent proteins and/or those immunolabeled with fluorescent antibodies.

More critically, a supplemental protocol introduced for the calibration and deformation correction of expanded tissue images would be beneficial when it enhances the precise measurement of structures in 3D. The current work includes a conceptual result using a photoactivatable dye integrated in a hydrogel with no biological sample, but the experiment in Figure S1F seems to have been carried out in 2D. In principle, two-photon lithography can form a 3D grid in a tissue-hydrogel hybrid. However, considering the necessary equipment and technical difficulties, particularly regarding how to secure a sufficient signal level by 3D lithography because the signal drop in Figure S1F is already remarkable after 1.5x expansion, the authors should suggest a practical 3D GelMap protocol with 3D deformation analysis and correction that users would be willing to choose instead of 3D calibration based on pre-expansion images.

➤ **Upon request of the reviewer, we have performed additional experiments with tissue labeled for total protein in combination with specific labeling of the pre- and post-synaptic proteins Bassoon and Homer, respectively. To validate the scaling and correction based on the GelMap grid, we have quantified the separation of these synaptic proteins and found the average synapse separation to be 117 ± 38 nm, consistent with previous reports (see references in manuscript). Interestingly, we also separately measured synapse separation of synapses oriented in the image plane versus synapses approximately oriented along the z-axis. We found no significant difference between the two values, suggesting that the expansion factor determined parallel to the image plane are a good proxy for the expansion along the z-axis.**

Following the editorial guidance, we have reduced the emphasis on the photoactivation approach and removed any claims about practical performance, but instead present it as a potential avenue to establish three-dimension deformation mapping in the future.

2, The manuscript only includes results generated with TREx. Since the authors conclude that GelMap is anticipated to be compatible with many existing and new expansion protocols, it would be helpful to demonstrate to the reader using other well-established protocols. In particular, the merit of using GelMap will be maximal when a sample is post-expansion immunolabeled and reference-free calibration is necessary.

MAP must be a representative method in this class, and a revised protocol from the same group (eMAP, a non-iterative protocol; Park J et al. Science Advances 2021) might be a good candidate to demonstrate this.

➤ **To examine the applicability of GelMap for other variants of Expansion Microscopy, we have now tested our approach using four commonly used ExM variants: pro-ExM, MAP, TREx, and pan-ExM, an iterative expansion approach. As such, we have also used different denaturation**

approaches (i.e. enzymatic digestion and thermal denaturation) and different anchoring approaches (i.e. AcX and FA/AA), as well as pre- and post-expansion labeling. The results are shown in Supplemental Figure 5 and reveal that GelMap can be readily used in a wide variety of different expansion approaches.

3. The authors included many rigorous and careful works on deformation analysis. I recommend a couple of additional analyses to further improve the manuscript. The effects of grid proteins on expansion factors and deformation (e.g., no effects, suppressing gross deformation, lowering the expansion factor, or causing deformation along grid lines) might be of interest, and such information will be considered when users decide to exclude the use of grids in their expansion workflow once they successfully establish the protocol due to GelMap.

The squareness and local expansion factor (Figs. 2A and S2A,B) must be determined not only by expansion-related deformations but also by an analysis process (e.g., corner detection precision in pixelated images). Please provide the baseline error levels observed before expansion.

- To address whether the GelMap grids affect the expansion factor, we measured the expansion factor for gels assembled on bare glass, on coated glass with patterned UV illumination (to create grids), and on coated glass with nonpatterned UV illumination (to burn away all proteins). As shown in Supplemental Figure 4B, the expansion factors are very similar in all conditions. Based on the correlative pre- and post-expansion imaging that we did we have no indication that deformations predominantly occur along grid lines.

We thank the reviewer for pointing out that multiple analysis steps during the correction procedure could affect our estimates for squareness and local expansion factors. To provide the baseline error levels, we have performed both analyses on pre-expansion images of GelMap grids that were resampled to contain the same number of pixels as representative post-expansion grids (Supplemental Figure 6D). This provided the baseline error level that we have now included in the graph in Figure 2B. Likewise, by registering two identical images using BigWarp, we have now also determined the baseline error level for the field deformation obtained from landmark-based registration (Figure 2F,G). In all cases the baseline errors are much smaller than the absolute errors introduced by expansion.

4. The downsides and limitations of using GelMap have not been properly discussed in the manuscript. By using a fluorescent grid, the number of fluorescence channels for microscopy should decrease by one. When an antibody is used to label grid proteins, the host species should be excluded from the antibody pool for immunolabeling of target proteins. It would be helpful for the reader if the additional steps and the cost of using GelMap for the initial setup and routine usage were summarized in the manuscript. The signal level of a grid is, again, of concern. In Figures S3A, 2C, and 4B,C, grid signals are found to noticeably drop after expansion, while other channels (e.g., tubulin) seem relatively stable. It would be helpful to provide a guideline for the necessary grid signal level for successful calibration.

- Setting up GelMap would require purchasing the equipment for photolithography (photomask, UV lamp, coverslip holder and vacuum pump), which costs approximately 10-12 k€, the price of a good objective. After this investment the costs of producing coverslips with GelMap grids are very low, just requiring coverslips and purified protein. We are currently looking into options to make GelMap grids commercially available, in which case the initial investments would not be required. The use of GelMap indeed requires a fluorescent channel, although due to the improved optical sectioning after expansion, we also found that the same channel can be used for specific labeling of subcellular structures, provided these structures (or the imaging of them) are positioned a bit deeper into the cell than the coverslip surface. An example of this is shown in Figure 4 where labeling of endogenous Bassoon is done in the same channel and using the same host species as the amplification of GelMap using the same secondary antibody. Finally, we anticipate that signal intensities after expansion can still be optimized by engineering additional labeling sites into the nanobody or through introduction of an expansion cassette (as in Real et al, Biorxiv 2023). In our revised manuscript, we have added these considerations to our discussion.

Minor points:

5. *Figure S2A,C: please indicate beside the color bars how the colors match the values.*

➤ **We have now added the value corresponding to the extremities of the color bars.**

6. *It is unclear how to interpret the data in Figure 2C,E–G and those in Figure S2D–G differently.*

➤ **The data in Supplemental Figure 2 is a replicate of the experiment shown in Figure 2. From comparing the orange, uncorrected curves it is apparent that the baseline deformations are different between these two examples, but that the corrective power of GelMap is similar.**

7. *Please improve the Materials and Methods section, which contains incomplete/unclear sentences, undefined acronyms, and missing catalog numbers.*

➤ **We apologize for any inaccuracies and have carefully checked and improved our Materials and Methods section.**

Decision Letter, first revision:

Dear Lukas,

Thank you for submitting your revised manuscript "GelMap: Intrinsic calibration and deformation mapping for expansion microscopy" (N METH-A51443A). It has now been seen by the original referees and their comments are below. The reviewers find that the paper has improved in revision, and therefore we'll be happy in principle to publish it in Nature Methods, pending minor revisions to satisfy the referees' final requests and to comply with our editorial and formatting guidelines.

In response to referee 3, we ask that you explicitly state any artifacts that can arise from doing 3D distortion corrections from a 2D measurement in the main text (probably Discussion).

TRANSPARENT PEER REVIEW

Nature Methods offers a transparent peer review option for new original research manuscripts submitted from 17th February 2021. We encourage increased transparency in peer review by publishing the reviewer comments, author rebuttal letters and editorial decision letters if the authors agree. Such peer review material is made available as a supplementary peer review file. Please state in the cover letter 'I wish to participate in transparent peer review' if you want to opt in, or 'I do not wish to participate in transparent peer review' if you don't. Failure to state your preference will result in delays in accepting your manuscript for publication.

ORCID

IMPORTANT: Non-corresponding authors do not have to link their ORCIDs but are encouraged to do so. Please note that it will not be possible to add/modify ORCIDs at proof. Thus, please let your co-authors

know that if they wish to have their ORCID added to the paper they must follow the procedure described in the following link prior to acceptance:

Sincerely,
Rita

Rita Strack, Ph.D.
Senior Editor
Nature Methods

Reviewer #1 (Remarks to the Author):

The revised manuscript fully addressed my concerns and provided improved technical details for prospective users. I believe the workflow using GelMAP would be benefit for the ExM and broader community.

Here are some small things to correct in the maintext:

1. Typo: Line 152, (Fig. 42A-B). "Figure", "Fig", "Fig." are all used in the text.
2. Reference missing: Line 68/Line 318 "Moreover, there are cases when pre- and post-expansion correlation is not possible, e.g. in the case of post-expansion labeling"
3. Fig. 2 dashed line in black overlaps with the axis, probably needs to change to another color

Reviewer #2 (Remarks to the Author):

I would like to compliment the authors on their work. They have addressed all the concerns and questions I raised in my initial review. Notably, the authors have successfully demonstrated the applicability of the GelMap approach to a wide range of expansion microscopy protocols. In addition, I thank the authors for including the supplementary figures describing the workflows. I highly recommend the publication of this work, which will undoubtedly be very useful to the scientific community.

Reviewer #3 (Remarks to the Author):

I appreciate the authors for their thoughtful responses with convincing results. The authors have addressed most of the concerns I initially had. Nevertheless, I request the authors to provide further clarification within the manuscript. I agree with the new claim that a precisely estimated expansion factor, derived from the 2D features on the GelMap grid, adequately accounts for the overall expansion occurring in 3D, as demonstrated with the bassoon and homer experiment. However, the manuscript omits to mention that GelMap only corrects for 2D deformation although deformation must occur in 3D. I believe that GelMap remains a valuable tool for expansion microscopy, even with its emphasis on 2D correction.

Author Rebuttal, first revision:

Point-to-point response for:***GelMap: Intrinsic calibration and deformation mapping for expansion microscopy****Hugo Damstra, Josiah Passmore et al.***Reviewer #1:**

The revised manuscript fully addressed my concerns and provided improved technical details for prospective users. I believe the workflow using GelMAP would be benefit for the ExM and broader community.

Here are some small things to correct in the maintext:

- 1. Typo: Line 152, (Fig. 42A-B). "Figure", "Fig", "Fig." are all used in the text.*
- 2. Reference missing: Line 68/Line 318 "Moreover, there are cases when pre- and post-expansion correlation is not possible, e.g. in the case of post-expansion labeling"*
- 3. Fig. 2 dashed line in black overlaps with the axis, probably needs to change to another color*

- **We thank the reviewer for the kind words. We have made the figure referencing consistent throughout the manuscript. We have clarified the text in lines 68 and 318 where we argue that correlation between pre- and post-expansion images is not possible when the structure of interest is labeled after expansion (i.e. there is no pre-expansion structure to image). We believe there is not one reference for this claim, as it is a conceptual point. Finally, we have changed the color of the dashed line indicating the baseline error in Fig 2.**

Reviewer #2:

I would like to compliment the authors on their work. They have addressed all the concerns and questions I raised in my initial review. Notably, the authors have successfully demonstrated the applicability of the GelMap approach to a wide range of expansion microscopy protocols. In addition, I thank the authors for including the supplementary figures describing the workflows. I highly recommend the publication of this work, which will undoubtedly be very useful to the scientific community.

- **We appreciate the kind words and feedback on our revised manuscript.**

Reviewer #3:

I appreciate the authors for their thoughtful responses with convincing results. The authors have addressed most of the concerns I initially had. Nevertheless, I request the authors to provide further clarification within the manuscript. I agree with the new claim that a precisely estimated expansion factor, derived from the 2D features on the GelMap grid, adequately accounts for the overall expansion occurring in 3D, as demonstrated with the bassoon and homer experiment. However, the manuscript omits to mention that GelMap only corrects for 2D deformation although deformation must occur in 3D. I believe that GelMap remains a valuable tool for expansion microscopy, even with its emphasis on 2D correction.

- **We thank the reviewer for the kind words. We have changed the section in the discussion to emphasize the reviewers point about correction in two dimensions and that it is unclear how this relates to deformations in the third dimension.**

Final Decision Letter:

Dear Lukas,

I am pleased to inform you that your Article, "GelMap: Intrinsic calibration and deformation mapping for expansion microscopy", has now been accepted for publication in Nature Methods. Your paper is tentatively scheduled for publication in our October print issue, and will be published online prior to that. The received and accepted dates will be Jan 12, 2023 and August 4, 2023. This note is intended to let you know what to expect from us over the next month or so, and to let you know where to address any further questions.

Over the next few weeks, your paper will be copyedited to ensure that it conforms to Nature Methods style. Once your paper is typeset, you will receive an email with a link to choose the appropriate publishing options for your paper and our Author Services team will be in touch regarding any additional information that may be required.

You will receive a link to your electronic proof via email with a request to make any corrections within 48 hours. If, when you receive your proof, you cannot meet this deadline, please inform us at rjsproduction@springernature.com immediately.

Please note that *Nature Methods* is a Transformative Journal (TJ). Authors may publish their research with us through the traditional subscription access route or make their paper immediately open access through payment of an article-processing charge (APC). Authors will not be required to make a final decision about access to their article until it has been accepted. [Find out more about Transformative Journals](https://www.springernature.com/gp/open-research/transformative-journals)

Authors may need to take specific actions to achieve [compliance](https://www.springernature.com/gp/open-research/funding/policy-compliance-faqs) with funder and institutional open access mandates. If your research is supported by a funder that requires immediate open access (e.g. according to [Plan S principles](https://www.springernature.com/gp/open-research/plan-s-compliance)) then you should select the gold OA route, and we will direct you to the compliant route where possible. For authors selecting the subscription publication route, the journal's standard licensing terms will need

to be accepted, including [self-archiving policies](https://www.springernature.com/gp/open-research/policies/journal-policies). Those licensing terms will supersede any other terms that the author or any third party may assert apply to any version of the manuscript.

Your paper will now be copyedited to ensure that it conforms to Nature Methods style. Once proofs are generated, they will be sent to you electronically and you will be asked to send a corrected version within 24 hours. It is extremely important that you let us know now whether you will be difficult to contact over the next month. If this is the case, we ask that you send us the contact information (email, phone and fax) of someone who will be able to check the proofs and deal with any last-minute problems.

If, when you receive your proof, you cannot meet the deadline, please inform us at rjsproduction@springernature.com immediately.

Once your manuscript is typeset and you have completed the appropriate grant of rights, you will receive a link to your electronic proof via email with a request to make any corrections within 48 hours. If, when you receive your proof, you cannot meet this deadline, please inform us at rjsproduction@springernature.com immediately.

Once your paper has been scheduled for online publication, the Nature press office will be in touch to confirm the details.

Once your paper has been scheduled for online publication, the Nature press office will be in touch to confirm the details.

Content is published online weekly on Mondays and Thursdays, and the embargo is set at 16:00 London time (GMT)/11:00 am US Eastern time (EST) on the day of publication. If you need to know the exact publication date or when the news embargo will be lifted, please contact our press office after you have submitted your proof corrections. Now is the time to inform your Public Relations or Press Office about your paper, as they might be interested in promoting its publication. This will allow them time to prepare an accurate and satisfactory press release. Include your manuscript tracking number NMETH-A51443B and the name of the journal, which they will need when they contact our office.

About one week before your paper is published online, we shall be distributing a press release to news organizations worldwide, which may include details of your work. We are happy for your institution or funding agency to prepare its own press release, but it must mention the embargo date and Nature Methods. Our Press Office will contact you closer to the time of publication, but if you or your Press Office have any inquiries in the meantime, please contact press@nature.com.

Nature Portfolio journals [encourage authors to share their step-by-step experimental protocols](https://www.nature.com/nature-research/editorial-policies/reporting-standards#protocols) on a protocol sharing platform of their choice. Nature Portfolio 's Protocol Exchange is a free-to-use and open resource for protocols; protocols deposited in Protocol Exchange are citable and can be linked from the published article. More details can found at www.nature.com/protocolexchange/about.

Best regards,
Rita

Rita Strack, Ph.D.
Senior Editor
Nature Methods